# Light-regulated growth from dynamic swollen substrates for making rough surfaces

Lulu Xue [1], Xinhong Xiong[1], Baiju P. Krishnan[1], Fatih Puza[1], Sheng Wang [1], Yijun Zheng[2 ✉] & Jiaxi Cui [1,3 ✉]

Natural organic structures form via a growth mode in which nutrients are absorbed, transported, and integrated. In contrast, synthetic architectures are constructed through fundamentally different methods, such as assembling, molding, cutting, and printing. Here, we report a photoinduced strategy for regulating the localized growth of microstructures from the surface of a swollen dynamic substrate, by coupling photolysis, photopolymerization, and transesterification together. Photolysis is used to generate dissociable ionic groups to enhance the swelling ability that drives nutrient solutions containing polymerizable components into the irradiated region, photopolymerization converts polymerizable components into polymers, and transesterification incorporates newly formed polymers into the original network structure. Such light-regulated growth is spatially controllable and dose-dependent and allows fine modulation of the size, composition, and mechanical properties of the grown structures. We also demonstrate the application of this process in the preparation of microstructures on a surface and the restoration of large-scale surface damage.

[1] INM - Leibniz Institute for New Materials, Campus D2 2, 66123 Saarbrücken, Germany. [2] School of Physical Science and Technology, ShanghaiTech University, Shanghai 201210, China. [3] Institute of Fundamental and Frontier Sciences, University of Electronic Science and Technology of China, Chengdu, Sichuan 610054, China. ✉email: zhengyj@shanghaitech.edu.cn; Jiaxi.Cui@leibniz-inm.de

L iving organisms are able to create various fascinating microstructures via a growth mode[1]. During the natural growth process, nutrients are absorbed into the body, transported inside and then integrated into the organisms under the directive of intrinsic code[2,3]. In contrast to this fully dynamic and open approach in nature, synthetic materials suffer from self-organized mechanisms to continuously incorporate external mass without compromising the material's integrity. In this regard, fundamentally different methods, such as assembling[4], molding[5], cutting[6], and printing[7,8], have been utilized and applied to fabricate man-made substances. Recently, applying the concept of growth to design self-organized synthetic systems has become a powerful strategy to develop novel dynamic materials with different biofunctions[9,10]. For instance, Gong et al. reported a kind of self-growing hydrogel that responds to repetitive mechanical stress through mechanochemical transduction[9]. In transduction, the supplied monomers are incorporated into the original polymer network by mechano-generated radicals to self-strengthen the materials. Additionally, Johnson et al. developed a class of growable polymer gels by using trithiocarbonate iniferters as dynamic connectors[10,11]. The iniferters can incorporate monomer molecules entrapped in the gels to elongate the polymer segments between crosslinked points. By applying a similar approach, Kloxin and coworkers have developed covalently crosslinked polymer networks in which crosslinking reactions can be triggered to strengthen the material or heal damage in the material[12]. These reported studies indicate that the growth strategy is promising for the postvariation of material properties. Despite the progress in this field, a growth strategy has not yet been applied to create microstructures on the surface, e.g., to enable localized growth of a structure from a flat substrate. To this end, a set of mechanisms for not only molecular incorporation but also mass transport and homogenization of polymer composition should be combined in a single system.

Many stimuli, such as light[13,14], strength[15,16], temperature[17], and moisture[18], have been applied to selectively trigger chemical reactions for spatial functionalization. Among them, light is environmentally friendly, noncontact, and spatiotemporally controllable and therefore is widely used for lithography[19], 3D printing[20], robotic actuation[21], cell migration[22], self-healing[23], switchable transitions[24], etc. It has also been employed to initiate the incorporation of entrapped monomer molecules into a polymer matrix for material growth[10–12,25]. However, in these previous examples, photoinduced reactions were utilized to convert monomers/crosslinkers into polymers rather than to guide mass transport.

Herein, we report a photoregulated strategy to control localized growth of microstructures from the surface of swollen substrates. In our design, three kinds of reactions, namely, photolysis, photopolymerization, and transesterification, were coupled together to guide the transport of liquid components entrapped in the substrates, to convert the polymerizable components in the liquids into polymers, and to reconfigure newly formed and original polymers. As a result of these reactions, microstructures can grow directly from flat substrates without the requirement for any preprogramming. The developed light-induced growth approach is spatially controllable, dose-dependent, and multitriggerable and can be used to create various rough surfaces or restore large-scale surface damage.

## Results

**Design and sample preparation.** The detailed concept of light-induced growth is demonstrated in Fig. 1. The growth starts from a swellable substrate composed of photoresponsive polymers crosslinked by ester-based linkers (Fig. 1a). The substrate can swell a solution consisting of monomer, crosslinker, photoinitiator, and transesterification catalyst, which is defined as the nutrient solution (Fig. 1b). Photolabile side groups are designed as promoters that can undergo photolytic reactions to generate dissociable ionic groups to enhance the swelling ability by expansion of the polymer networks to transport the nutrient solution into the irradiated region (Fig. 1c)[26]. This photoinduced mass transport is coupled with the photopolymerization of the monomer and crosslinker in the nutrient solution, leading to continuous swelling of the irradiation region and thus the formation of protrusions on the irradiated surface (Fig. 1d). During swelling, the original networks are stretched, which should prevent the nutrient solution from diffusing. A transesterification catalyst is thus designed to trigger an exchange reaction between original and newly formed networks to release such mechanical tension and to reconfigure the grown structure (Fig. 1e). We expect that such a coupling of three reactions could lead to an on-demand, localized growth of microstructures from material surfaces.

A material system of 4-hydroxybutyl acrylate (HBA), o-nitrobenzyl acrylate (NBA, Supplementary Fig. 1), and 1,6-hexanediol diacrylate (HDDA) was selected to demonstrate the design. HBA is a commercially available precursor for making polymer substrates with good monomer swelling ability, while NBA is a photolabile monomer (promoter) that can generate dissociable ionic –COO⁻ groups to induce an increase in swelling ability[27,28]. The HDDA crosslinker has an ester linkage that can undergo a transesterification reaction with the hydroxyl group of HBA. Poly (HBA-co-NBA) samples with different NBA molar fractions (0, 5, 10, 20, 35, and 50%) were fabricated via photopolymerization under blue LED light (460 nm, 0.6 mW cm⁻²) in the presence of Irgacure 819 (I-819, initiator). Under this irradiation condition, the NBA unit will not undergo photolysis since it does not absorb at this wavelength. This hypothesis was verified by irradiating an NBA solution under the same conditions (no visible change in either UV-Vis or ¹H NMR spectroscopy, Supplementary Fig. 2). After photopolymerization, unreacted components were washed with ethanol, and the obtained specimens were denoted as seed-x, where x is the feeding molar fraction of the NBA. Both Fourier transform infrared (FTIR) spectroscopy and UV-Vis spectroscopy were used to characterize the chemical structure of the seeds. Typical IR signals assigned to the -NO₂ group of NBA units (1528 cm⁻¹: asym. NO₂ stretch; 1342 cm⁻¹: sym. NO₂ stretch) and strong UV absorption from the nitrobenzyl group were observed (Supplementary Fig. 3), indicating the successful copolymerization of o-nitrobenzyl ester units.

The seeds showed an excellent swelling ability to the nutrient solution consisting of HBA (monomer), HDDA (crosslinker), I-819 (photoinitiator), and benzensulfonic acid (BZSA, transesterification catalyst). For example, an equilibrium swelling ratio of 4.6 was obtained for a seed-20% sample with a thickness of 1.4 mm (Supplementary Fig. 4). The polymerizable components, i.e., HBA and HDDA, entrapped in the polymer networks could undergo photopolymerization to integrate into the seeds under UV light irradiation (365 nm, 10 mW cm⁻², confirmed by the weight of the sample after removal of unreacted components by washing). UV light was also expected to trigger the photolytic reaction of the NBA unit (confirmed by the disappearance of –NO₂ peaks in the FTIR spectrum, Supplementary Fig. 5). It was noted that under our irradiation conditions, UV light could not trigger obvious chain scission of polymer networks except photolysis of NBA (Supplementary Fig. 6). In addition, the thermal effect generated by polymerization triggers transesterification reactions to release any polymerization-induced mechanical tension in such dynamic networks[29–31].

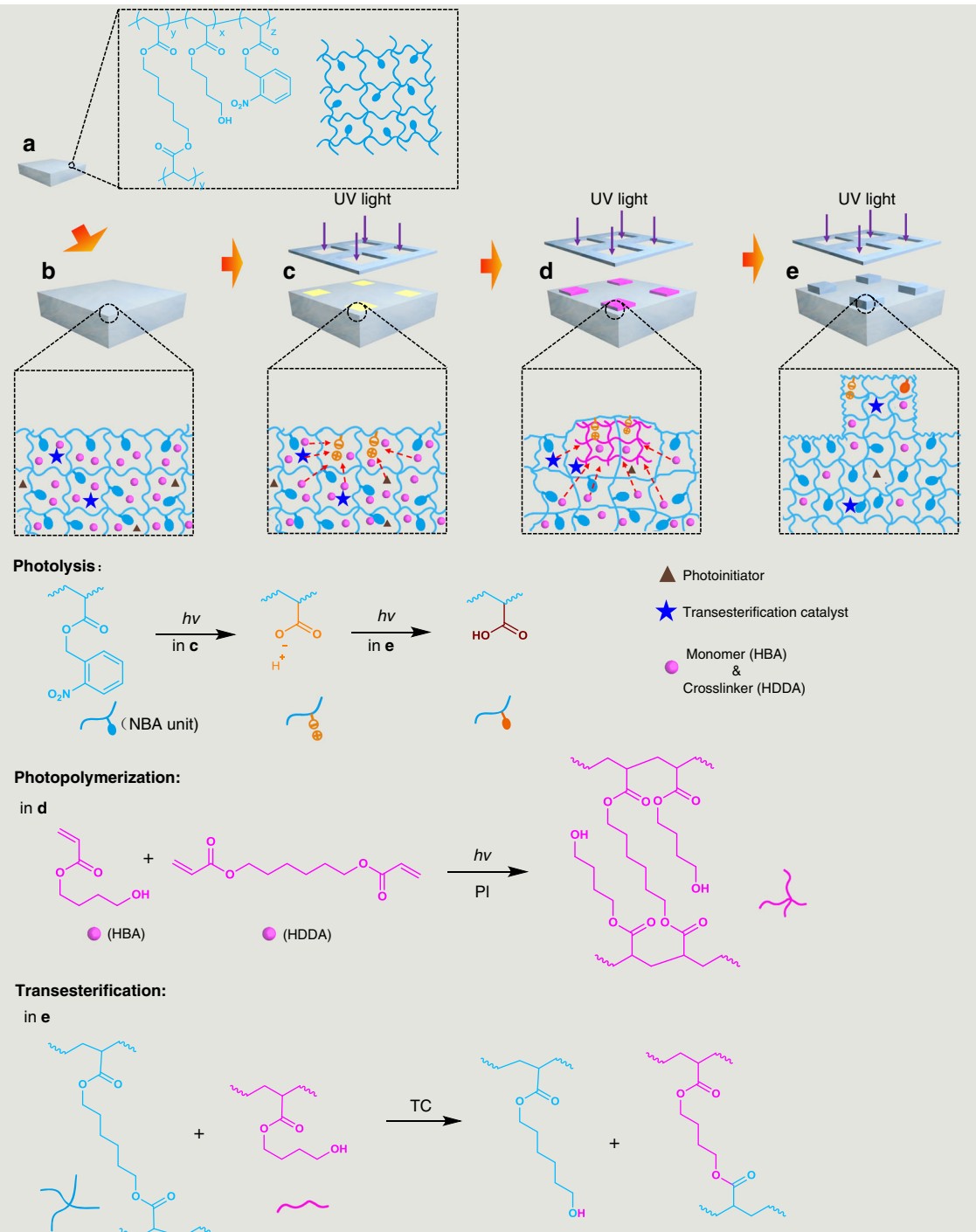

**Fig. 1 Schematic of light-induced growth from swollen substrates. a** Growable seed made from 4-hydroxybutyl acrylate (HBA), o-nitrobenzyl acrylate (NBA, promoter), Irgacure 819 (I-819, photoinitiator) and 1,6-hexanediol diacrylate (HDDA). **b** Swollen seed. The mixture of HBA, HDDA, photoinitiator (I-819), and transesterification catalyst (benzensulfonic acid (BZSA)) were used as the nutrient solution for swelling. **c** Swollen substrate under selective UV irradiation. Photolysis of NBA units generated dissociable ionic groups to induce liquid diffusion into the irradiated region. **d** New polymer network formed via photopolymerization. Liquid components diffused in, and the polymer chains in the original network were stretched. **e** The grown part was homogenized via transesterification reactions between the original and newly formed polymer networks.

**Light-triggered localized growth**. Figure 2a shows an image of light-induced growth of a pillar from the surface of a flat swollen seed-20%. During UV irradiation, a regular structure slowly grew out from the irradiated region of the material surface, and the height of this pillar could reach up to 250 μm during the testing time.

Figure 2b shows the growth process evaluated by the height of the grown pillar. The growth process of swollen seed-20% was triggered by irradiation with either 365 nm or 460 nm light. Exposure to 365 nm light triggered both polymerization and photolytic reactions, while 460 nm light induced only polymerization. In this regard, the experiment with 460 nm light could be

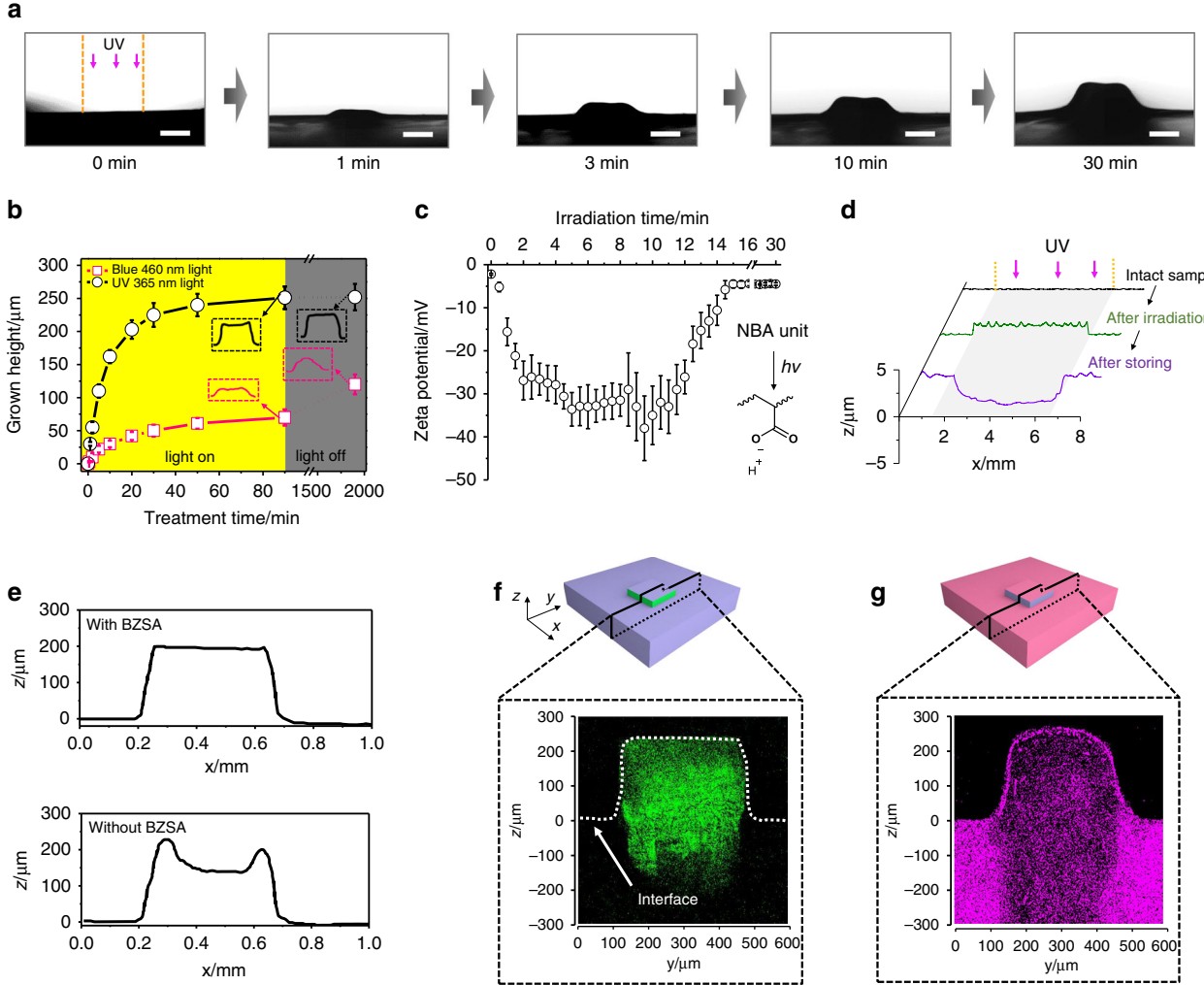

**Fig. 2 Light-regulated growth from HBA-based substrates. a** Time-dependent images (side view) of swollen substrates under UV irradiation (10 mW cm$^{-2}$). A photomask with a diameter of 500 µm was used. The scale bar is 250 µm. **b** The height of the grown microstructures (grown height) of different samples vs treatment time. The data were obtained from three independent measurements. Error bars are s.e.m. The dashed boxes show the typical profiles of the grown structures before and after being stored in the dark. **c** Zeta potential of linear poly(HBA-*co*-NBA) copolymers at different irradiation times. The polymer concentration was 2 mg mL$^{-1}$, and an LED lamp (10 mW cm$^{-2}$) was used for irradiation. The inset shows the photolytic reaction of the NBA units. **d** Profiles of a swollen seed-20% containing HB acetate, I-819, and BZSA under different conditions. Photomasks with a diameter of 5 mm were used. **e** Typical profiles of the grown structures obtained from the swollen seed with (top) or without (bottom) transesterification catalyst BZSA. Fluorescent cross-section images of the grown structure obtained from a nondyed seed and dyed nutrient (**f**) and a dyed seed and nondyed nutrient (**g**). The substrates used in (**a**), (**d**), (**e**), (**f**), and (**g**) contain 20% promoter (seed-20%). PDIDA was used to dye the seed in (**f**) and the nutrient in (**g**) with a concentration of 0.01 wt%.

used as a control to evaluate the contribution of the photolytic reaction. The irradiation intensity of both lights was the same (10 mW cm$^{-2}$), and such a design was used to achieve similar photopolymerization effects (photopolymerization conversions reached their plateaus in 2 min, Supplementary Fig. 7). Upon irradiation with 365 nm light, the height of the grown structure increases rapidly in the first 5 min (75 µm) and reaches a plateau at a value of 250 µm in 50 min. The grown sample retains its shape after being stored in the dark overnight. In comparison, growth also occurs in the control sample, but the height of the grown structure at the plateau is significantly smaller (70 µm). We attributed the growth in the absence of a photolytic reaction to the fact that photopolymerization consumed the monomer and crosslinker in the nutrient solution to form new polymer networks in the irradiated region, followed by creation of a concentration gradient of the monomer and crosslinker to drive these components to diffuse into the irradiated region to participate in the polymerization. Such a polymerization-diffusion cycle led to

the growth of the structure. However, this photolysis-absent growth is significantly slower than the photolysis-present growth, and the obtained structure is also notably smaller. The higher grown structure obtained in photolysis-present growth indicated that the photolytic reaction had generated an extra effect to accelerate the diffusion of the nutrient solution into the irradiated region. Since the photolytic reaction of NBA units generates carboxyl groups, we attributed the extra effect to the formation of dissociable ionic –COO$^{-}$ groups, which enhanced the swelling ability of the irradiated region by expansion of the polymer networks via electrostatic repulsion. In addition to having a smaller size, the grown structure is seriously distorted after being stored in the dark (Fig. 2b and Supplementary Fig. 8).

A growth process based on the polymerization-diffusion cycle was thus proposed: upon UV irradiation, photolysis of NBA units and polymerization of the monomer and crosslinker in the irradiated region generated the dissociable ionic group of –COO$^{-}$ and the concentration gradient of monomer and crosslinker,

which significantly enhanced the swelling ability of the irradiated region. As a result of the enhanced swelling ability and lower concentrations of the monomer and crosslinker in the irradiated region, the nutrient solution diffused into the irradiated region to induce a polymerization-diffusion cycle. In this cycle, photo-polymerization was significantly faster than the diffusion process (time for photopolymerization to reach its conversion plateau: ~2 min; time for liquid molecules to diffuse into the seed to fully swell it: ~4 h without irradiation and ~2 h under irradiation; see Supplementary Figs. 7 and 9 for more detail). After irradiation, the generated concentration gradient still existed and continued driving the monomer and crosslinker (major liquid component of the nutrient solution) to diffuse into the grown structure to swell. This swelling distorted the grown structure of the photolysis-absent sample (swollen state, Supplementary Fig. 10). To confirm this, after the distorted grown structures were washed, consider-able shrinkage was observed in the distorted sample. In addition, the obtained grown structures were stable in both the swollen and dried states (Supplementary Fig. 11).

We studied the formation of the dissociable ionic group of –COO⁻ by monitoring the zeta potential of a linear poly(HBA-co-NBA) containing 20% NBA units ($Mn$ = 8500, $PDI$ = 1.16, Supplementary Figs. 12 and 13) under UV illumination. As shown in Fig. 2c, the copolymer is almost neutral (−1.8 mV) before irradiation. The potential value decreases sharply to −27 mV after 2 min of UV irradiation, indicating the formation of dense negative species on the copolymer (–COO⁻). The species was assigned as a carboxyl ion, which was released from the NBA unit[28]. It was highly active and was ultimately neutralized into –COOH (thus a change from −32 mV to −2 mV in the zeta potential). To verify that the change in the zeta potential were caused by the dissociable ionic –COO⁻ group rather than the photolytic product of the o-nitrobenzyl moiety, we selected 2-nitrobenzyl alcohol as a control, a compound that can undergo a similar photolytic reaction but does not generate a carboxyl group. The zeta potential of the 2-nitrobenzyl alcohol solution did not change obviously under UV irradiation (Supplementary Fig. 14), indicating that the change in zeta potential of poly(HBA-co-NBA) should be attributed to the generation of dissociable –COO⁻. The formation of –COO⁻ on the polymer segments induced an increase in swelling ability[26], while the carboxyl groups reduced the swelling ability of the matrix to reduce the distortion effect. We proved this hypothesis by irradiating a control sample of seed-20% swollen by a nonpolymerizable solution consisting of 4-hydroxybutyl acetate (HB acetate, Supplementary Fig. 1), I-819, and BZSA. Upon UV irradiation, a bulge forms on the irradiated region (Fig. 2d, Supplementary Fig. 15), indicating a swelling process. Since the liquid compositions were nonpoly-merizable (lacking a polymerization-diffusion cycle to drive liquid diffusion), the driving force for swelling was attributed to charge-induced electrostatic repulsion. Although a change of only ~5 μm in height was observed, such a change in the polymerization-diffusion cycle could be amplified. To further confirm the contribution of photolysis to accelerate mass transport, we compared the diffusion rate of the nutrient solution in seed-20% under different conditions by a swelling method (Supple-mentary Fig. 16). The average rate of mass transport under irradiation conditions was significantly higher ($4.7 \times 10^{-5}$ cm² s⁻¹) than that without irradiation ($4.9 \times 10^{-6}$ cm² s⁻¹). After the transition of –COO⁻ to –COOH, the liquid composites diffused from the irradiated region, resulting in a cave surface. This result was consistent with the fact that irradiated seed-20% showed lower swelling ability into the nutrient solution because of the formation of carboxyl side groups (Supplementary Fig. 4). The decrease in swelling ability favored the formation of nondistorted

grown structures since the transport of the nutrient solution from the nonirradiated region to the irradiated region during storage was reduced (Fig. 2b).

Transesterification indeed occurred during light-induced growth. Under our irradiation conditions, the temperature of the irradiated region in a swollen seed-20% sample increased to 62 °C in the initial 1 min (Supplementary Fig. 17), while that of swollen seed-20% with nonpolymerizable liquids was unchanged even after 60 min UV irradiation (Supplementary Fig. 18). At this temperature, the catalyst BZSA used in our system can induce efficient transesterification[31]. We studied the contribution of transesterification to the grown structure by a control sample without BZSA. In contrast to the flat surface of the grown structure observed in the typical sample, the surface of the grown structure obtained from the control sample was concave (Fig. 2e, Supplementary Fig. 19). During growth, since photopolymeriza-tion of the monomer and crosslinker in nutrient solution was significantly faster than the transport of nutrient solution, the growth rate was mainly dependent on the diffusion rate of the nutrient solution. In the control sample, the nutrient solution diffused into the irradiated region from outside and was integrated into the periphery via rapid polymerization; thus, fewer monomer and crosslinker molecules could diffuse and be integrated into the center, leading to an energy-unfavorable concave surface. In the presence of a transesterification catalyst, such an energy-unfavorable concave surface could be converted into an energy-favorable flat surface via transesterification-associated reconfiguration (Fig. 1e and Supplementary Figs. 20 and 21)[29]. The transesterification was further proven by the significantly higher modulus of the grown structure of the control samples (Supplementary Fig. 22). Without transesterification, a double-network structure formed to stiffen the grown structure (490 KPa), while transesterification-induced homogenization reduced this stiffening effect (380 KPa)[32].

The composition of the grown structure was studied by confocal fluorescence spectroscopy. To enable imaging and detailed investigation of the swollen substrates, the nutrient solution was labeled by a fluorescent crosslinker, bis-N,N'-6-hydroxyhexanol perylenetetracarboxylic diimide-acrylate (PDIDA, Supplementary Fig. 1). This crosslinker was stable under our irradiation conditions (Supplementary Fig. 23), and its diacrylate structure was expected to significantly decrease its relocalization possibility during transesterification-induced homogenization. Seed-20% was soaked in a nutrient solution containing 0.01 wt% PDIDA, followed by light-induced growth for 30 min. After polymerization, the unreacted components were washed with ethanol/CHCl₃ solutions before being subjected to confocal imaging. Figure 2f shows the cross-section of the grown samples. Compared to the dark surroundings, a bright color was observed in the grown part, indicating that the monomer and crosslinker in the nutrient solution were integrated into the grown structure. The bright color extends to the bottom region of the seed, suggesting that the growth started inside the sample rather than simple polymerization from the surface of the sample. Complementary experiments were also conducted to assure the growth mechanism. Seed-20% was dyed with 0.01 wt% PDIDA and then grown from a nondyed nutrient solution (Fig. 2g). As expected, the grown region was still fluorescent but significantly diluted, indicating that the grown region was made from both original and newly formed networks. The fluorescence intensity of the surface of a newly grown structure was nearly the same as that observed in the nonirradiated region (Supplementary Fig. 24), indicating that nearly no growth occurred in the surface layer. This might be due to the evaporation of monomer molecules in this region or the lower swelling ability of the surface layer of the sample. In addition, the fluorescence intensity gradually increased

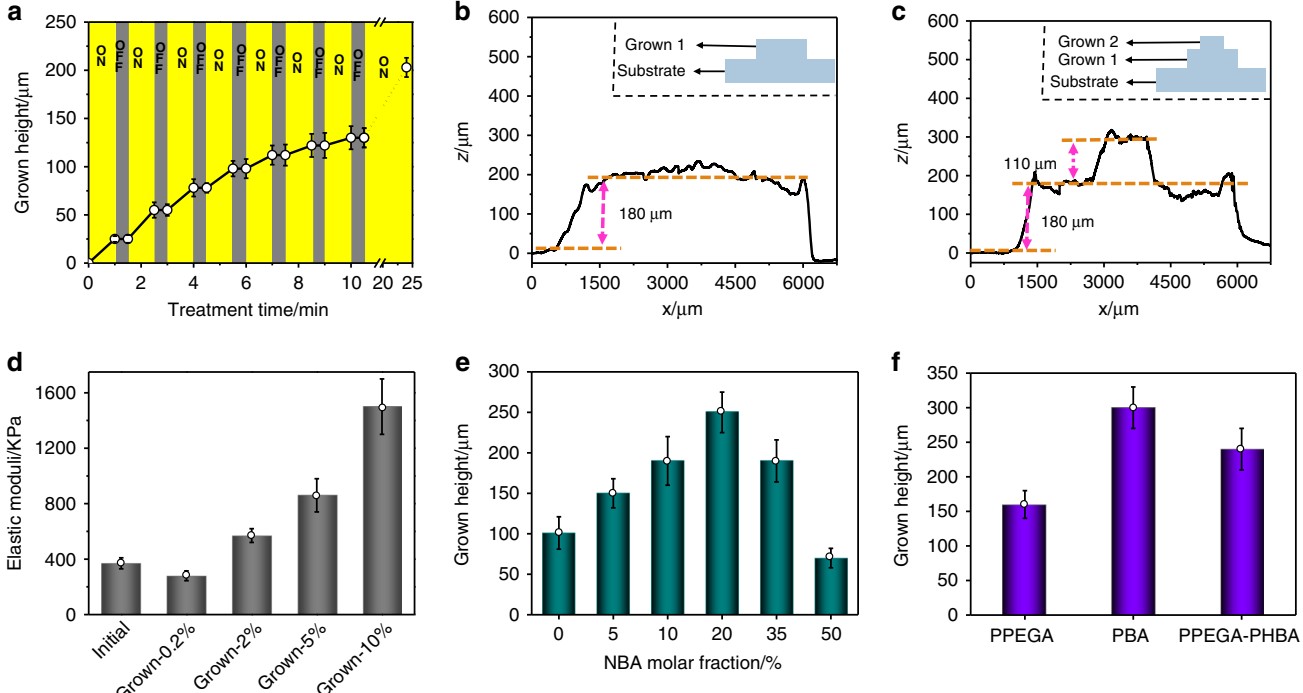

**Fig. 3 Control of light-induced growth. a** The height of the grown structures changes with irradiation conditions. The labels "ON" and "OFF" indicate the state of UV light applied to the samples. **b** The profile of the grown structure obtained from first-cycle growth. A round photomask with a diameter of 5000 μm was used. **c** The profile of the grown structure prepared from two-cycle growth. The grown sample in (**b**) was used, and it was reswelled in nutrient solution for growth. A round photomask with a diameter of 1250 μm was used, and irradiation was selectively applied to the grown region. **d** Modulus of the seed and the grown structures obtained from the nutrient solution with different crosslinker concentrations (x in Grown-x in the label). **e** The height of the grown structure obtained from the seed containing different NBA molar fractions. **f** The height of the grown structure made from different monomers. PPEGA/PBA: PEGA/BA-based seeds grew from PEGA/BA-based nutrient solution; PPEGA-PHBA: PEGA-based seeds grew from HBA-based nutrient solution. Seed-20% was used in (**a**), (**b**), (**d**), and (**f**), and the data in (**a**), (**d**), (**e**), and (**f**) were obtained from three independent measurements. Error bars are s.e.m. The height values in (**e**) and (**f**) were collected at plateau states.

from the top region to the bottom region. We attributed this finding to the gradual swelling of the networks. Based on the growth curves (Fig. 2b), the growth rate decreased with time due to consumption of the monomer. Therefore, the dilution effect in fluorescence decreased from top (early stage) to bottom (later stage) in the growth direction.

**Control of growth**. The light-induced growth was not only localized (Supplementary Figs. 25 and 26) but also temporally controllable. We employed seed-20% to demonstrate this capability by switching the irradiation light. As shown in Fig. 3a, growth was triggered only by the implementation of irradiation. For example, the height of the grown structures increased to 25 μm within the first min of activation, and the growth stopped when light was turned off. The growth was reinitiated by turning the light on. Such on-off modulation can extend until the growth reaches its plateau. In addition, several parameters, including the crosslinking degree of the seed, the diameter of the irradiation region, and the light intensity, were studied to modulate growth. Increasing the crosslinking degree of seed-20% decreases its swelling ability as well as the height of the grown structure at the plateau state (Supplementary Fig. 27). Upon increasing the irradiation diameter, the growth height at the plateau state increased in the range from 266 μm to 600 μm but decreased in the range of >600 μm (Supplementary Fig. 28). We attributed the increase to the photopolymerization-induced thermal effect (elevation in temperature would accelerate the diffusion rate of liquid molecules and thus the growth). Considering thermal dissipation,

increasing the irradiation diameter favored temperature elevation. On the other hand, increasing the diameter also elongated the diffusion distance and thus reduced the growth. This reducing effect became more obvious in the larger diameter range (>600 μm). For a decrease in light intensity reduced the growth because of slower photolysis and polymerization reactions (Supplementary Fig. 29).

Sequential growth was also possible. Figure 3b shows a sample with a grown pillar with a height of 180 μm and a diameter of 5000 μm. This grown sample can be swelled by the nutrient solution again and activated to grow in the grown region. Under the same irradiation conditions, a new pillar with a height of 110 μm formed on the previously grown pillar (Fig. 3c). The lower height of second growth was attributed to the lower concentration of the promoter in the second photolysis step.

Since the grown structure was made from feed nutrient solutions and original polymers, its composition could be easily regulated by the nutrient solutions, which provided a powerful approach to control the mechanical properties of the grown structure. We demonstrated this concept by varying the crosslinker fraction in the nutrient solutions used to seed-20% (made from 1 wt% crosslinker, modulus: 370 KPa). When a nutrient solution with a crosslinker concentration of 0.2 wt% was used, a grown structure with a modulus of 280 KPa was obtained. On the other hand, increasing the crosslinker concentration in the nutrient solutions enhanced the modulus of the grown structure. The E-moduli were even up to 1.5 MPa when a crosslinker concentration of 10 wt% was used in the nutrient solution (Fig. 3d). Notably, such a growth method to spatiotemporally

change the modulus of the grown structures did not induce any interface issue since the newly formed structure was grown from inside the original materials.

The promoter fraction in the seed was also expected to be an important parameter to control growth. In principle, increasing its fraction should enhance the driving force for liquid transport into the irradiation region but would also decrease the final swelling ratio of the irradiation region since both the NBA unit and its photolytic product reduced the material's swellability to nutrient solutions. As shown in Fig. 3e, the height of the grown pillar in the plateau state increases with the fraction of promoters in the range of <20% but decreases in the range of 35–50%.

The concept of photoinduced growth could be applied to different material systems. We demonstrated this applicability with poly(ethylene glycol) methyl ether acrylate (PEGA), a hydrophilic monomer, and butyl acrylate (BA), a hydrophobic monomer (Supplementary Figs. 30 and 31). Figure 3f lists the growth heights of different material systems under the same growth conditions. The height of PEGA grown in the plateau state (160 μm) is lower than that of HBA (250 μm). This was attributed to the significantly higher viscosity of PEGA (90 cSt, 20 °C)[33] than of HBA (10.7 cSt, 20 °C). The higher viscosity led to a lower transport rate and thus less growth. The hypothesis was supported by the higher height (300 μm) of the grown pillar made from BA, which has a lower viscosity (0.92 cSt, 20 °C). Moreover,

a hybrid system could also be created by varying the compositions of the nutrient solution. For example, we grew PEGA-based seeds in HBA-based nutrient solution. The grown pillar reached a height of up to 240 μm and showed a modulus of 580 KPa when the seed had a modulus of 220 KPa (Supplementary Fig. 32). Based on these results, we concluded that the light-induced growth was fully controllable and allowed for fine variation in size, strength, and composition.

**Application demonstration.** Localized growth of microstructures from a flat substrate implied a template-free method for making a patterning surface. Figure 4a–e shows a tentative example. Upon UV illumination through a mask, a regular micropattern (diameter of 500 μm) grew from the flat surface of the swollen sample (Fig. 4a–d). The formed pillars were uniform, with a height of 250 μm (Fig. 4e). In addition, direct writing with a UV laser was also possible (Supplementary Fig. 33)

The growth could be used to restore large-scale surface damage at the millimeter level. Self-healing of extensive damage is extremely challenging since it not only involves molecular reconfiguration but also requires significant mass transport[34]. Although dynamic materials have been suggested to be self-healable, they mainly depend on the rebonding of matrices, which is normally useful in the recovery of microcracks and scratches[35]. It is difficult for them to restore large-scale surface damage at the millimeter level[36]. Since

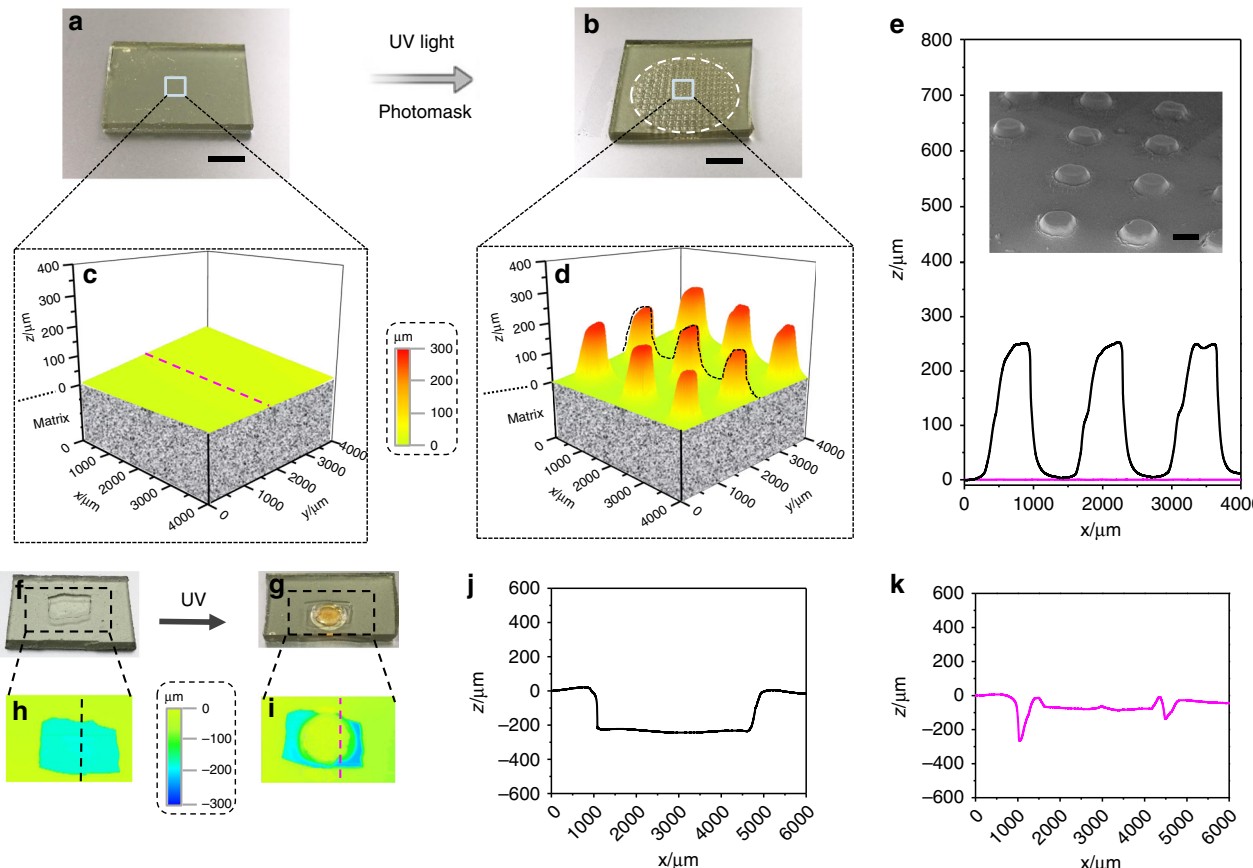

**Fig. 4 Application demonstration of light-induced growth. a–e** Microstructure pattern grown from a flat substrate (scale bar: 8 mm): **a** swollen seed-20%; **b** formed microstructure. The dashed line highlights the irradiated zone. 3D profiles of the surfaces of the swollen seed-20% (**c**) and the grown sample (**d**). **e** Surface profiles of the seed−20% and the formed microstructure. The regions are highlighted by dotted lines in (**c**) and (**d**). The inset in **e** shows the SEM image of the growth pattern. Scale bar: 500 μm. **f–k** Restoration of large-scale surface damage by light-induced growth: **f** images of the swollen seed-20% with damage of 0.60 cm (*l*) × 0.40 cm (*w*) × 0.22 mm (*h*) then treated with UV light (**g**). Top view of the 3D profile of the damaged swollen seed-20% (**h**) and the grown swollen seed-20% (**i**). Surface profiles of the damaged area before (**j**) and after (**k**) healing. A photomask was used for UV irradiation, and the irradiation time was 30 min. The scale shows the height. The dashed lines in (**h**) and (**i**) highlight the positions used for the collection of the profile data.

light-induced growth involves significant liquid transport, in situ polymerization, and reconfiguration, we assumed that it could be used to restore large-scale surface damage by guiding growth toward the damaged region. The promising demonstration of surface damage restore by the developed strategy is detailed in Fig. 4f–k. Damage with a size of 0.60 cm ($l$) × 0.40 cm ($w$) × 0.22 mm ($h$) was tentatively created on a substrate made from seed-20%. For a better comparison, we partially irradiated the damaged region and induced growth until it was flush with the undamaged part. It could be observed from the profile of the damaged zone that the irradiated region was regenerated, implying a powerful method to restore large-scale surface damage.

## Discussion

We have demonstrated a strategy for designing photoinduced growable materials. The strategy is based on coupling three kinds of reactions to achieve localized growth: photolysis to generate dissociable ionic groups to increase the swelling ability and drive the diffusion of a nutrient solution into the irradiated region, photopolymerization to convert the monomer and crosslinker in the nutrient solution into crosslinked polymers, and transesterification to homogenize the newly formed and original polymer networks. Such light-induced growth is spatially controllable and dose-dependent and allows fine modulation of the size, composition, and mechanical properties of the grown structure. The flexible tunability enables the creation of microstructures on surfaces and the restoration of large-scale surface damage. Although the methodology developed in this study was demonstrated on structured surfaces, the mechanistic insights gained regarding governing growth can be readily applied to change the bulk properties of materials in consideration of the capability of light to spatially trigger various reactions. We thus envision that its development will benefit areas such as self-healing materials and rough surfaces.

## Methods

**Chemicals and materials**. 4-Hydroxybutyl acrylate (HBA) (TCI Deutschland GmbH), butyl acrylate (BA) (99%, Sigma-Aldrich) and poly(ethylene glycol) methyl ether acrylate (PEGA) (average $Mn = 480$ g mol$^{-1}$, Sigma-Aldrich) were purified by passing through a column of neutral alumina to remove the inhibitors before being used. 2-Nitrobenzyl bromide (98%, Alfa Aesar), acrylic acid (99%, Sigma-Aldrich), potassium carbonate (K$_2$CO$_3$) (99%, Alfa Aesar), 1,4-butanediol (99%, Fluka), acetic acid (99%, ABCR), bis(2,4,6-trimethylbenzoyl)-phenylphosphineoxide (I-819) (Ciba), 1,6-hexanediol diacrylate (HDDA) (99%, Alfa Aesar), sulfuric acid (H$_2$SO$_4$) (95-98%, Sigma-Aldrich), benzenesulfonic acid (BZSA) (98%, Sigma-Aldrich), 2-nitrobenzyl alcohol (97%, Sigma-Aldrich), 3,4,9,10-perylenetracarboxylic dianhydride (98%, Alfa-Aesar), acrylic chloride (97%, Sigma-Aldrich), imidazole (ACS reagent, Sigma-Aldrich), triethylamine (Et$_3$N) (99.5%, Sigma-Aldrich), sodium chloride (99.9%, ABCR), sodium carbonate (Na$_2$CO$_3$) (99%, Sigma-Aldrich), sodium sulfate (Na$_2$SO$_4$) (99%, Sigma-Aldrich), 4-Cyano-4-(phenylcarbonothioylthio)pentanoic acid (CPADB, Sigma-Aldrich), and 6-aminohexanol (95%, TCI Deutschland GmbH) were used as received. N,N-dimethylformamid (DMF) (99.8%, anhydrous, Sigma-Aldrich), dichloromethane (DCM) (99.8%, anhydrous, Sigma-Aldrich) and chloroform (99.8%, anhydrous, Sigma-Aldrich) were used directly. Other solvents like petroleum ether and ethyl acetate were purchased from ABCR and used without any treatment. 2,2′-Azobisisobutyronitrile (AIBN, 98%, Sigma-Aldrich) was purified by recrystallization from ethanol.

**Instruments**. $^1$H NMR and $^{13}$C NMR spectroscopy of the products were obtained with a Bruker 300 MHz nuclear magnetic resonance equipment using CDCl$_3$ and DMSO-$_{d6}$ as solvents. Mass spectra were carried out on an Agilent LC/MSD SL. The number-average molecular weight ($Mn$) and polydispersity index ($Mw/Mn$, PDI) of polymers were measured by a Agilent HPC 1100 gel permeation chromatography (GPC) system using a PSS-GRAM pre-column with a series of PMMA as standard samples. Ultraviolet-visible (UV-vis) spectroscopy were obtained from a Varian Cary 4000 UV-visible spectrophotometer. ESEM images were captured on a FEI ESEM Quanta 400 FEG. Attenuated total reflection-Fourier transform infrared (ATR-FTIR) spectroscopy were recorded with a Bruker VERTEX 70v FTIR spectrometer. Fluorescent spectroscopy were conducted with a Hitachi F-7000 fluorescence spectrophotometer. Infrared camera (InfraTec GmbH, Germany) with VarioCAM HD head was used for recording the temperatures during the polymerization. Surface profile and 3D profile of the specimens were carried out on a SURFCOM 1500SD3. Optical microscope images were acquired from a Nikon ECLIPSE LV100ND. Zeta potential of linear polymers was obtained with a Malvern Zetasizer Nano ZSP. Fluorescent images were recorded on a LSM 880 confocal, and ImageJ software was used to analyses the data. Primo experiments were conducted on a Total Internal Reflection Microscopy. Water contact angles of materials were collected by a OCA 20 instrument. Side view of self-growing microstructures on material surfaces was recorded in the OCA 20 machine. Column chromatography was performed using silica gel (215–400 mesh). UV 365 nm and blue 460 nm collimated LED light (Olympus BX & 1X, 1700 mA) was provided by THORLABS, of which intensity was set as 10 mW cm$^{-2}$ during the experiments. Blue light LED strip lamp (460 nm) with an intensity of 0.6 mW cm$^{-2}$ was obtained from amazon online and used to initiate the polymerizations. Elastic moduli of seed samples were measured on a universal testing machine (ZWICK 1446, Germany) with a load cell of 200 N and crosshead velocity of 10 mm min$^{-1}$ and values were calculated in the linear elastic region of the stress-strain curves from 1 to 5%. Every measurement was conducted three times. The elastic moduli of seed-0% and seed-20% samples were measured by compression test with a load cell of 2 KN and velocity of 2 mm min$^{-1}$. The values were calculated in the linear elastic region of the stress-strain curves from 0.1 to 0.5%. The elastic moduli of the growing structures were obtained by indentation experiments. The ASMEC indenter type is Berkovich equipped with a diamond tip. Samples were struck on the PEEK model before measurements. Indentations were carried out in the load-controlled mode, with an initial quadratic up to 20 mN within 10 s, a creep period of 5 s, and a quadratic decrease of the force to 0.08 mN within 5 s. The results were collected by eight different areas for each sample and analyzed according to the Fast hardness and modulus measurements (ISO 14557). The fit range of the unloading curve is from 98 to 40%.

**Fabrication of seeds**. Taking seed-20% as an example: to a mixture of HBA (80 mol%) and NBA (20 mol%) were added HDDA (crosslinker, 1 wt%) and I-819 (photoinitiator, 1 wt%) to obtain the precursor. The precursor solution was coated on Teflon substrates and cured under blue light (intensity: 0.6 mW cm$^{-2}$) for 20 min. The obtained substrate was immersed in ethanol, and the solution was changed every 8 h (3 times) to remove unreacted specimens. Then, it was dried to afford seed-20%.

**Light-induced growth**. Seed-20% was immersed in a nutrient solution containing HBA (96 wt%), HDDA (1 wt%), I-819 (1 wt%) and BZSA (2 wt%) for 12 h to obtain swollen samples. For growth, the swollen samples were subjected to UV light (intensity: 10 mW cm$^{-2}$) with a suitable photomask.

## Data availability

All data used for this paper are available from the authors on request.

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

## Acknowledgements

This work was supported by BMBF under the project of the Leibniz Research Cluster "Organic/synthetic multifunctional microproduction units—New ways to the development of the active ingredient" with award number 031A360D. Y.Z. acknowledge the support of starting funding of ShanghaiTech University while L.X. and X.X. acknowledge the support from the Chinese Scholarship Council (CSC).

## Author contributions

J.C. and Y. Z. conceived the concepts of the research. J.C. supervised the research. L.X. and J.C. designed the experiments. L.X., X.X., B.K., F.P., and S.W. performed the experiments. L.X., X.X., Y.Z. and J.C. analyzed the results; L.X., Y.Z. and J.C. wrote the paper. All authors commented on the paper.

## Competing interests

The authors declare no competing interests.

## Additional information

9