## [Peer Review File · Nature Communications]

Reviewers' comments:

Reviewer #1 (Remarks to the Author):

This work is on a very hot topic of self-growth of materials. The authors reported a Light-regulated site-specific growth of microstructures from swollen elastomers. They claimed that the material growth in this work shares analogy to the growth of living organisms. This manuscript is well written, and the results are interesting. This work provides a useful method for surface modification or surface pattern of gel materials. While the flaw of this method is that, it is difficult to change to bulk property of materials. The authors should indicate the limitation of their method more clearly. In addition, the reviewer is not fully convinced by the explanation on the growth process. Before considering for publication, the following questions or concerns should be addressed.

1) The role of NBA unit on growth kinetics. The authors claimed that the photolabile NBA unit can generate a charge pair of carboxyl ionic and hydron to raise high osmotic pressure, which drives the nutrient solution to transport into the irradiated region for polymerization. They compared the growth kinetics of materials with and without NBA unit. First, I want to say, this comparison is not fair, as 20% molar ratio of NBA gives a totally different polymer network structure compared to that without of NBA, which should influence the growth kinetics. Second, the authors claimed that the size of new grown structure without NBA is remarkably smaller than that with NBA. This is not true at long treatment time. As shown in Figure 2b, the size of new grown structure without NBA is even much larger than that with NBA around 2000 min.

2) The authors attributed the growth of sample without NBA to the diffusion-polymerization cycle and claimed that the growth kinetics (growth kinetics (~ 50 min)) is significantly faster than the diffusion kinetics of the nutrient (~ 12 hours). How did they get these two characteristic timescales? From Figure 2b it seems to be impossible to get these two timescales as growth and diffusion are coupled together.

3) How about the long-term stability of the new grown structure? For application, one night is obviously not enough. Did the authors have the long-term stability data on the new grown structure? The authors said that the new grown structure from the seed without NBA distorts after stored in dark. It is better to put an image or data to show this distortion. While it is still unclear to the reviewer why the diffusion of liquid components after polymerization leads to the distortion of grown structure. The reviewer also does agree with the explanation "the formation of acrylic acid from the promoter decreases the swelling ratio of the growth region (Supplementary Fig. 5)". The acrylic acid is small molecules, which diffuse into surrounding solution very fast in the experiment of Supplementary Fig. 5. It seems to me that the photolysis gives a smaller side chain length, leading to a smaller swelling ratio of seed.

4) To demonstrate the photo-induced charges of NBA units, the authors measured Zeta potential of poly(HBA-co-NBA) solutions. Is it possible to measure the Zeta potential of seed sample, as the Zeta potentials of the linear polymer and crosslinked network may be different?

5) The data in Figure 2d seems not support the authors' argument that the osmotic pressure from charges is important for liquid transport. In Figure 2d, the size change is less than $5 \mu\text{m}$, which is far smaller than the size of new grown structure ($\sim 250 \mu\text{m}$). In Figure 2b, the seed without NBA has a larger size of new grown structure than that with NBA. This also suggest that the NBA is not so important as the authors argued. On the opposite, the Figure 2b and 2d suggest that the diffusion may play the dominant role on the size of new grown structure.

6) The size profiles of grown structure with and without transesterification catalyst (BSA) are interesting. For the sample without BSA, when the concave forms? And how it changes with time? Is it due to edge effect? as the chemicals diffuse much easier in the edge than in the center of grown structure.

7) The idea of fluorescent crosslinker is quite clever. From Figure 2g, we can see that the surface of new grown structure has brightest color, which is almost same as the seed part. If it is due to the transesterification, I think the color should gradually change from bright to dark in the growth direction. To me, it seems that the UV light leads to the local swollen of irradiated region. At same time, the polymerization also happens in this region. Then the polymerization-diffusion cycle leads

to the gradual swollen of the network and the new polymerization always happens inside the original network. This process gives the fluorescent images of Figure 2f and 2g.

8) In Figure S14, the growth height shows a nonmonotonic change with diameter of photomask. Do the authors have some explanations? In Figure 15, did the growth reach equilibrium when comparing the effect of light energy?

Reviewer #2 (Remarks to the Author):

Micro and nano patterns are very important to surface chemistry and physics, and their fabrication technology is being attracted much attention. This manuscript from Lulu Xue, et al reports a direct photo-induced strategy to control site-specific growth of microstructure from swollen elastomers. Microstructures can grow directly from flat substrates without any preprogramming by guiding continuous molecule transport, interconversion, and reconfiguration sequentially. This is innovative and meaningful for the formation of pattern with spatially controllable, dose-dependent. I suggest a revision of the manuscript before it can be accepted for publication. Some comments shown as following.

1. How about the minimum resolution of pattern can be fabricated by this method? Is it possible to produce nano or submicrometer scale pattern.
2. In Figure 2(g), the fluorescence weakening in the grown region may be caused by fluorescence quenching by UV-irradiation for long time, rather than significantly diffusion. How to eliminate the factors of fluorescence quenching in this experiment?
3. In Figure 2(b), why does the grown height of seed-0% suddenly increase even exceed the height of seed-20% after light off, while the height of seed-20% remains unchanged?
4. Page 7, it is mentioned that "after the charge disappears, the liquid composites diffuse out from the irradiated region, resulting in a cave surface", It is better to give more comments on this point.
5. Can the average rate of mass transport be calculated? The specific value?
6. In this manuscript, it is mentioned that the grown sample retains its shape after stored in dark overnight, how long will it last, and whether there will be collapse or deformation?
7. How about the quantitative relationship between the expansion rate or cross-linking degree of substrate and the growth height?

Reviewer #3 (Remarks to the Author):

The manuscript presents an elaborate system for inducing spatially and temporally controlled, light-triggered polymerization resulting in a geometrically restricted growth of a polymer sample. The authors propose that the design is based "on the coupling of three kinds of reactions together, including photolysis to generate a driving force for liquid transport, photopolymerization to incorporate the liquid, and transesterification to homogenize the newborn component".

While the subject matter of the paper is interesting and the amount of the data presented is substantial, the manuscript requires major revision, in order for it to be considered for publication in a high-impact outlet, such as Nature Communications.

There are substantial deficiencies in the manuscript, in terms of both the scientific substance, (presented experimental evidence, its interpretation, and conclusions drawn) and the quality of its presentation.

The paper claims that due to the somewhat elaborate design presented (that is the combination of photolysis, liquid transport, polymerization, and transesterification), the authors have developed a "strategy for designing photo-induced growing materials" with "potential applications in

transformable materials, additive manufacture, self-healing materials, actuating systems, adaptive materials, etc.” This claim is very broad, while the system presented is quite narrow in scope. The presented strategy could nonetheless be considered a development worthy of publication, but one would still have to ask themselves in what level of a journal. While the level of novelty of this work is debatable and whether it merits publication in a Nature family journal is a matter of opinion, there are several elements in the manuscript that this referee finds to be deficiencies significant enough that without correcting them, the paper cannot be considered for publication.

Authors reference a very relevant work by J. Johnson et al (ref. 10: Chen, M. et al. Living additive manufacturing: transformation of parent gels into diversely functionalized daughter gels made possible by visible light photoredox catalysis. *ACS Cent. Sci.* 3, 124-134 (2017)), but neglect to reference other critically important works from the same group: Zhou, H. X.; Johnson, J. A. Photo-controlled Growth of Telechelic Polymers and End-linked Polymer Gels. *Angew. Chem., Int. Ed.* 2013, 52, 2235–2238; Gordon, M. B.; French, J. M.; Wagner, N. J.; Kloxin, C. J. Dynamic Bonds in Covalently Crosslinked Polymer Networks for Photoactivated Strengthening and Healing. *Adv. Mater.* 2015, 27, 8007–8010. This is a serious oversight, because all these papers, taken together, demonstrate that the use of PHOTOACTIVATED steps for growth and post-polymerization modification of polymeric materials for the very similar to this paper’s purposes have significant precedent in the literature.

In addition to many problems with the language/grammar used (some of which are listed further down in this review), there are as many instances where the terminology used is simply wrong, often so wrong that it precludes understanding what the authors actually meant. This applies to chemical terminology, materials science terminology, and layman’s language alike. Some examples from the first several pages of the manuscript are below.

Abstract:

“The strategy is based on the coupling of three kinds of reactions together, including photolysis to generate a driving force for liquid transport, photopolymerization to incorporate the liquid, and transesterification to homogenize the newborn component.” It is debatable whether photolysis, as such, generates a driving force for liquid transport (see additional discussion of this issue further below). Photopolymerization (or any other polymerization) does not incorporate a liquid, it is a chemical reaction that incorporates a monomer and/or crosslinker into the polymer.

Transesterification does not homogenize the newborn component; it results (in this case) in a redistribution of ester linkages within the polymer network(s). And “newborn component” needs a clear definition, in order to be used in a scientific paper.

“We demonstrate its (“light-regulated self-growth”) applications in creating structural surfaces and restoring large damage.” What is meant by “structural surfaces”? What is meant by “large damage”?

P. 2: “Johnson et al. developed a class of growable polymer gels containing an initiator of reversible addition-fragmentation chain-transfer polymerization (RAFT) as a connector.” The sentence utilizes chemical terminology, but one cannot refer to RAFT polymerization as “a connector”.

P. 2: “Many strategies have been developed to selectively trigger chemical reactions for spatial functionalization, including light, strength, temperature, moisture, etc.” None of the words used (light, strength, temperature, moisture) can be described as a “strategy”.

P. 3: “Herein, we report a photo-regulated strategy to control site-specific growth of microstructure from swollen elastomers.” “Site-specific” is not a good word in this context. Likewise, “growth of microstructure from swollen elastomers” is difficult to understand for a

polymer scientist and a general reader alike.

P. 3: "In our design, three kinds of reactions, including photolysis, photopolymerization, and transesterification, are coupled together to guide continuously molecule transport, interconversion, and reconfiguration sequentially." Even if one can accept (with difficulty) the term "molecule transport", "molecule ... interconversion and reconfiguration" does not carry much meaning, and "continuously" and "sequentially" only additionally muddle the intended message.

P. 4: "while NBA is a photolabile monomer (promoter) that can generate a charge pair of carboxyl ionic and hydrion, to raise high osmotic pressure." Probably "carboxyl ion" (not "ionic") is meant, but even if so, what does "to raise high osmotic pressure" mean? Also, I do not remember when last time I saw the term "hydrion" in a scientific/chemistry paper.

As stated above, this is just a handful of examples from the first several pages, but there are many more throughout the text. Such misuse of scientific and non-scientific terminology precludes, in many cases, understanding of what the authors actually meant to say, which is unacceptable for a scientific publication in any journal.

Important scientific issues to consider:

1) One of the central messages the authors are making is about the role of photolysis in inducing (stimulating or accelerating, rather) liquid transport from the bulk of the swollen samples into the areas where irradiation occurred. The authors attribute it to the increase in osmotic pressure. This is a key aspect, by which they distinguish their work from a very carefully done work coming from groups of Matyjaszewski, Kowalewski and Balazs (ref. 23 in the manuscript: Cuthbert, J. et al. Transformable materials: structurally tailored and engineered macromolecular (STEM) gels by controlled radical polymerization. *Macromolecules* 51 ,3808-3817 (2018)). To support their statement, the authors cite ref. 26 (Alexander, S. et al. Charge renormalization, osmotic-pressure, and bulk modulus of colloidal crystals-theory. *J. Chem. Phys.* 80 , 5776-5781 (1984)). This work, however, applies to a substantially different system and if the authors somehow conclude that there is direct relevance between the physical treatment of colloidal crystals presented in this work and their case of inducing faster/more pronounced liquid influx into the irradiated areas of the polymer network, they should explicitly describe and justify the connection. Or explicitly state that the high osmotic pressure argument they are making is a mere hypothesis. It is also quite important to note that the infiltrating liquid in question is not very polar, so one has to reconcile that with the proposed increase in driving force for it to diffuse into the irradiated regions containing charge pairs. There may be other explanations, or hypotheses, compatible with the experimental observations. For example, there are mechanical properties/diffusion/crosslinking arguments to consider.

2) Also, related to this, the attribution by the authors of the less pronounced swelling after sample irradiation (with the promoter present) to the formation of acrylic acid (Fig. S5) is not sufficiently substantiated at all. The evidence of the water contact angle decrease (Fig. S4 a, b) is indirect, while what is claimed as the FTIR evidence of the appearance of carboxyl group (Fig. S4c) and the text accompanying it are less than convincing (both in the substance and in the language used: Section 6 of SI:

"Water contact angle was used to monitor the photolysis of the substrates. It changes from 95o to 63o after UV irradiation to generate acrylic side group (Figure S4a and 4b). FTIR shows the disappearance of the -NO₂ after UV illumination as the peak at 1528 cm⁻¹ of OH group became broader (Figure S4c).

The photolysis of NBA units to generate acrylic side groups decreases the swelling ratio of the seeds to the nutrient solution. Figure S5 shows the swelling kinetics of the photolytic film (1.4 mm) and the equilibrium swelling ratio decreases from 4.6 to 3.5."

With the authors' text and punctuation kept, it seems that the peak at 1528 cm⁻¹ is attributed to

OH and it has become broader. This is total confusion. The 1528 cm⁻¹ peak is assigned by the authors to the NO₂ group and the broadening of the OH group signal is i) extremely difficult to see and ii) even if it is there, it is not even a weak evidence of the appearance of a carboxyl group. Also, what is "acrylic side groups"??

Also, though the assignments of the IR bands due to the NO₂ group are likely correct, the IR spectrum of the polymer lacking the nitrobenzene group is needed/desired, in order to give more weight to this assignment.

3) P. 5: The authors state "In company with the polymerization, thermal effect generated by polymerization would trigger transesterification reactions to release any polymerization-induced mechanical tension in such dynamic networks.^{27, 28}"

There is no evidence presented of the temperature increase due to the polymerization. For this statement to be made, an IR-camera measurements are necessary. Moreover, the references 27 and 28 invoke temperatures of 130 C and higher for the transesterification reactions as the redistribution mechanism in the polymer networks they describe. So, here, as in the case of the increased osmotic pressure explanation (hypothesis, rather), the used references do not offer direct support to the statements made by the authors.

4) Fig. 2b:

What is the explanation of the seemingly very high growth of the control sample (the last point of the magenta curve)? The accompanying text (big part of P.6) leaves, again, a strong impression of unsubstantiated hypothesizing, especially when coupled with a frivolous use of several different tenses, as the excerpt below shows.

"Compared to this, the control also shows a growth in the irradiated region. It was explained (WHEN?? BY WHOM??) by the fact that photopolymerization consumes the nutrient solution to form new polymer networks in the irradiated region, which would create a nutrient gradient to drive the liquid compositions to diffuse into the irradiated region to join the polymerization. Such diffusion-polymerization cycle thus led to the growth. However, this promoter-free growth is significantly slower than the promoter-presenting growth and the obtained structure is also remarkably smaller. Moreover, the grown structure seriously distorts after stored in dark. We attributed this to the fact that the growth kinetics (~ 50 min) is significantly faster than the diffusion kinetics of the nutrient (~ 12 hours)."

To use the term "kinetics" (rather than "time scale", for example), in the absence of kinetic measurements, and to measure it in minutes and hours is another example of careless use of scientific terminology.

5) If/when the above comments to the substance are taken into account and a major revision to the paper is made, as importantly, the English grammar and word usage of the text will need significant work. The examples are too numerous to list. Here is just a selection:

"...following by transport inside..." – followed by...

"fundamental different"

"fabricate artificial substances"

"since the lack..." (grammar)

"to apply growing method to create different structure"

"among them light takes advantages of environment friend, noncontact..."

"to character its photostability"

"a seed exhibit"

"a regularly structure tardily grow up from..."

"growing kinetics"

"benzensulfonic acid" – benzenesulfonic acid

"Irgacure" - Irgacure

Also, BSA is a widely used abbreviation that means "bovine serum albumin". Here it is used to mean Benzenesulfonic acid; maybe to avoid this abbreviation altogether?

In conclusion, though the material presented in the manuscript is interesting and it can attract a multidisciplinary audience, in the current form, the manuscript reads more as a poorly written collection of data lacking careful and well-supported discussion and interpretation. To claim the novelty, as it is claimed now, the distinctions of the current work from the previously published relevant works need to be made in a scientifically more profound and clear way. Where there is not enough experimental proof and substantiation from published data, the presented explanations should be left as hypotheses. Without such a major revision, the manuscript cannot be recommended for publication, in the opinion of this reviewer.

Point-by-Point Response to Referees

Reviewer #1:

This work is on a very hot topic of self-growth of materials. The authors reported a Light-regulated site-specific growth of microstructures from swollen elastomers. They claimed that the material growth in this work shares analogy to the growth of living organisms. This manuscript is well written, and the results are interesting. This work provides a useful method for surface modification or surface pattern of gel materials. While the flaw of this method is that, it is difficult to change to bulk property of materials. The authors should indicate the limitation of their method more clearly. In addition, the reviewer is not fully convinced by the explanation on the growth process. Before considering for publication, the following questions or concerns should be addressed.

Response: Thank you for these positive comments! The manuscript has been carefully revised according to the suggestions.

While the flaw of this method is that, it is difficult to change to bulk property of materials. The authors should indicate the limitation of their method more clearly.

Response: In this manuscript, we emphasize the capability of our growing method in creating microstructure on material surfaces. In principle, this method can also be applied to change the bulk property of materials by using two-photon technology to spatially trigger the growth inside a sample, since the promotor (*o*-nitrobenzyl group) is two-photon responsive. In the case that a two-photon initiator is used, it is possible to precisely modulate the bulk properties of a material. We have discussed this potential in the revised manuscript and the systematic investigation in this topic is on-going in our group.

“Although the methodology developed in this study was demonstrated on structuring surface, the mechanistic insights gained in governing the growth can be readily applied to change the bulk property of materials in consideration of the capability of light to spatially trigger various reactions.”

In addition, the reviewer is not fully convinced by the explanation on the growth process.

Response: We have discussed the growth process in more detail based on additional data and believe the revised version is convincing enough.

1) The role of NBA unit on growth kinetics. The authors claimed that the photolabile NBA unit can generate a charge pair of carboxyl ionic and hydron to raise high osmotic pressure, which drives the nutrient solution to transport into the irradiated region for polymerization. They compared the growth kinetics of materials with and without NBA unit. First, I want to say, this comparison is not fair, as 20% molar ratio of NBA gives a totally different polymer network structure compared to that without of NBA, which should influence the growth kinetics. Second, the authors claimed that the size of new grown structure without NBA is remarkably smaller than that with NBA. This is not true at long treatment time. As shown in Figure 2b, the size of new

grown structure without NBA is even much larger than that with NBA around 2000 min.

Response: Thank you for this insightful comment! We agree that it is not fair to compare the NBA-free system to the NBA-contained one. Therefore, we have conducted additional control experiments by varying the irradiation light for comparison (the same samples were used for comparison, Figure R1a and Figure 2b in the manuscript). The lights with wavelengths of 365 and 460 nm were used, respectively. The 365 nm light can trigger both polymerization and photolysis reactions (defined as photolysis-present condition) while the 460 nm light only induces polymerization (defined as photolysis-absent condition). Under our irradiation conditions (intensity of 10 mW/cm² for both 365 nm and 460 nm lights), both lights can result in similar photopolymerization effect (confirmed by control experiments in which the polymerization conversions reach their plateaus in 2 min; the polymerization processes are significantly faster than the growth, and compared to the growth, the polymerization processes can be considered as similar, Figure R1b and Figure S5 in the revised manuscript). When applied to the samples, the 460 nm light induces a slow growth and the final height of new grown structures only reached 70 μm (note that 460 nm can even induce slightly faster polymerization, Figure R1b), in comparison to the height of 250 μm obtained by 365 nm light irradiation (Figure R1a).

Figure R1. (a) Growth curve v.s. activation time of swollen seed-20% under 460 nm or 365 nm light irradiation. (b) Polymerization conversion of HBA/NBA/HDDA/I-819 under different irradiation times.

Experiment of measuring the polymerization kinetic: A mixture of HBA (80% molar ratio), NBA (20% molar ratio), HDDA (1 wt%), and I-819 (1 wt%) were mixed and purged with Argon for 20 min. 50 μL of the mixture (weighted as M_0) was taken for each polymerization with different exposure times (10 s, 30 s, 1 min, 2 min, 5 min, 10 min, 20 min, 30 min, 40 min, 50 min) under 460 nm or 365 nm light irradiation (intensity: 10 mW/cm²). After polymerization, the unreacted components was removed by ethanol rinsing (3 times), followed by drying to obtain the crosslinked polymers (weighted as M_t). The polymerization conversion was defined as M_t/M_0 .

As for the higher grown structure observed in the NBA-free sample, we emphasize that the structure become bigger during storage in dark because liquid compositions (nutrient solution) diffuse into the grown structure, rather than because of growth. Note that photopolymerization consume the monomer and crosslinker (liquid compositions) in the irradiated region, inducing a concentration

gradient which drives liquid compositions to diffuse into the region (and then further polymerization). Such polymerization-diffusion cycle leads to the growth of the structure. In this cycle, photopolymerization is significantly faster than diffusion (a complete photopolymerization under our irradiation condition is ~ 2 min while the diffusion time is ~ 4 hours—detailed calculation is discussed in the answer to Question 2). Therefore, after the irradiation, generated concentration gradient still exists and continues driving liquid compositions (monomer and crosslinker) to diffuse into the grown structure to make the structure become bigger (swollen state). The obtained structure is irregular and brittle. To confirm this, we washed the grown structure and observed a remarkable shrinkage (washed state, Figure R2a). Such irregular structure is undesirable. In the NBA-contained system, carboxyl side chains would be generated (from NBA units) to reduce such diffusion effect and the obtained structure is regular (see more detail in the answer to Question 3).

In the revised manuscript, we replaced the results of NBA-free sample with that of photolysis-absent sample which also shows similar distortion (Figure R2b and Figure S8 in the revised manuscript). We have also integrated the discussion in the revised manuscript to make our claim more convincing and clearer.

Figure R2. Profile of the grown structures of (a) seed-0% and (b) photolysis-absent sample under different conditions. Swollen state: sample after stored in dark for 32 h, washed state: sample after washing by ethanol.

2) The authors attributed the growth of sample without NBA to the diffusion-polymerization cycle and claimed that the growth kinetics (growth kinetics (~ 50 min) is significantly faster than the diffusion kinetics of the nutrient (~ 12 hours). How did they get these two characteristic timescales? From Figure 2b it seems to be impossible to get these two timescales as growth and diffusion are coupled together.

Response: Sorry for the misleading presentation. Here we want to emphasize that the polymerization of monomer and crosslinker is significantly faster than their diffusion. Although polymerization and diffusion are coupled together, the growth reach a plateau when most of monomer and crosslinker around the irradiated region have been consumed. The time to reach this plateau is ~ 50 min, which is defined as growth time (“growth kinetics”). After storage, liquid

compositions diffuse from remote region into the grown structure. The time for this long-distance diffusion is significantly longer than the growth time. We evaluated the time scale for polymerization and diffusion by control experiments. Please see the experiment detail for the polymerization in the answer for Question 1 (Figure R1b). Under our irradiation condition, the polymerization conversion reaches its plateau in 2 min. A swelling experiment was used to evaluate the diffusion time. Since the growth structure has a diameter of 500 μm , we assumed that diffusion time for liquid molecules to diffuse into the growth structure is similar to the time for a sample with a thickness of 500 μm to be completely swollen. In Figure R3 (Figure S7 in the revised manuscript), the time for the sample to reach complete swelling is nearly 4 hours without irradiation (tested in both nutrient solution and its non-polymerizable solution) but 2 hours under irradiation (tested in non-polymerizable solution).

We have modified our presentation to make it clear.

Figure R3. Swelling ratio of seed-20% thin film (thickness: 500 μm) in the solution containing HB acetate, I-819 and BZSA.

3) How about the long-term stability of the new grown structure? For application, one night is obviously not enough. Did the authors have the long-term stability data on the new grown structure? The authors said that the new grown structure from the seed without NBA distorts after stored in dark. It is better to put an image or data to show this distortion. While it is still unclear to the reviewer why the diffusion of liquid components after polymerization leads to the distortion of grown structure. The reviewer also does agree with the explanation “the formation of acrylic acid from the promoter decreases the swelling ratio of the growth region (Supplementary Fig. 5)”. The acrylic acid is small molecules, which diffuse into surrounding solution very fast in the experiment of Supplementary Fig. 5. It seems to me that the photolysis gives a smaller side chain length, leading to a smaller swelling ratio of seed.

Response: The new grown structure from seed-20% is stable. We evaluated the stability of the new grown structure in both swollen and washed states (washing treatment is used to remove unreacted monomer and crosslinker, Figure R4 and Figure S9 in the revised manuscript). After stored for 32 h, the grown structure of swollen sample become slightly bigger because liquid diffuses into the grown structure as a result of concentration gradient (note that the change is significantly smaller than that observed in NBA-free sample as shown in Figure R2), which still retained this shape for

1 month in dark. In the washed sample, the grown structure shows nearly same profile even after stored in dark for over 1 months.

Figure R4. Profile of grown structure from swollen seed-20% with (a) and without (b) washing by ethanol before and after stored in dark for 32 h and 1 month. Cylinder structures with diameter of $500 \mu\text{m}$ could grow up to $250 \mu\text{m}$ from swollen seed-20%, then one was put in the dark directly, the other was washed by ethanol then put in the dark.

We have collected the image to show the distortion after storage (Figure R5 and Figure S6 in the revised manuscript). The distortion observed after storage is attributed to the transport of liquid compositions from the unirradiated region into the grown structure. We assume that the reviewer is curious why such distortion occurs only in the storage step, rather than the irradiation (growth) step, since both steps involve in remarkable liquid transport. We attribute this to the photopolymerization-induced transesterification reaction. Photopolymerization elevates system temperature and trigger the transesterification which release any mechanical tension generated by liquid transport. In the storage step without polymerization, transesterification become extremely slow. Thus, the mechanical tension induced by liquid transport distort the structure. However, in the NBA-contained sample, the formation of carboxyl side groups decreases the swelling ability of the grown structure to the liquid and thus reduce this diffusion effect. As a result, the grown structure in NBA-contained sample retain its shape.

Figure R5. Images of grown structures of swollen seed-0% in as-prepared (a) and stored (b) states. The sample in (b) was stored in the dark for 32 h.

The statement of “the formation of acrylic acid from the promoter” is not appropriate. Sorry for this mistake! We have replaced “acrylic acid” with “carboxyl side groups” in our revised manuscript.

4) To demonstrate the photo-induced charges of NBA units, the authors measured Zeta potential of poly(HBA-co-NBA) solutions. Is it possible to measure the Zeta potential of seed sample, as the Zeta potentials of the linear polymer and crosslinked network may be different?

Response: We agree that it will be more convincing to show Zeta potential of seed sample. However, it is theoretically difficult to measure the Zeta potential of our sample since it is non-conductive. Note that the signal collected for evaluating the Zeta potential is the electric charge on electrodes. Even a Zeta potential analyzer used for solid sample was applied, the result reveals the potential of material’s surface, rather than bulk property. In this case, a control with a polymer solution in which the ions is mobile can reveal more convincing information.

5) The data in Figure 2d seems not support the authors’ argument that the osmotic pressure from charges is important for liquid transport. In Figure 2d, the size change is less than 5 μm , which is far smaller than the size of new grown structure ($\sim 250 \mu\text{m}$). In Figure 2b, the seed without NBA has a larger size of new grown structure than that with NBA. This also suggest that the NBA is not so important as the authors argued. On the opposite, the Figure 2b and 2d suggest that the diffusion may play the dominant role on the size of new grown structure.

Response: NBA is indeed important for getting the regular grown structure. It not only generates ion pair ($-\text{COO}^-\text{H}^+$) to enhance osmotic pressure to drive liquid to diffuse into the irradiation region during growth (irradiation) but also convert into carboxyl side group to reduce liquid diffusion into the new grown structure to distort the structure. Note that the growth is a polymerization-diffusion cycle. The liquid diffused into the irradiated region would be fixed in the region *via* photopolymerization and then induced a high concentration gradient to drive liquid to diffuse into the region. A self-accelerated polymerization-diffusion cycle is triggered. Since the polymerization is fast, the growth rate depends on the diffusion. The diffusion rate depends on the driving force. We believe that both concentration gradient and osmotic pressure contribute to the driving force. The comparison in Figure R1 indicate the contribution of the photolysis reaction of NBA for the growth. The data in Figure 2d was used to further prove that photolysis of NBA can indeed generate force to drive liquid to diffuse into the irradiation region. Therefore, NON-polymerizable liquid was used in this experiment. Although only a change of $\sim 5 \mu\text{m}$ was observed, such difference can be amplified in the polymerization-diffusion cycle. In addition, polymerization elevates the local temperature and the diffusion is also accelerated in growth. Therefore, it is meaningless to compare the results in Figure 2b and 2d.

Please see our explanation in the answers to questions 1 and 3 for the contribution of NBA to the growth.

6) The size profiles of grown structure with and without transesterification catalyst (BSA) are interesting. For the sample without BSA, when the concave forms? And how it changes with time? Is it due to edge effect? as the chemicals diffuse much easier in the edge than in the center of grown structure.

Response: We have conducted additional experiments to monitor the change in surface morphology of the sample without the catalyst BZSA (Figure R6 and Figure S15 in the revised manuscript). The concave forms at very beginning of the growth. We attribute it to faster polymerization process than diffusion during light-induced growth, rather than simple edge effect (from the viewpoint of mechanical tension, the edge effect should lead to a convex). The liquid diffused from unirradiated region will be triggered to undergo polymerization in periphery, thus less liquid components (monomer and crosslinker) can diffuse and integrated in the center than in edge. In other words, the pathway to the periphery is shorter than that to the center. Growth in the periphery is larger than that in center and therefore, edge is higher.

Based on the result observed in Figure 2f and 2g, the growth occurs inside the sample, rather than on the surface of the sample. We do not have any data to support that the diffusion in the edge is easier than that in the center of grown structure.

Figure R6. (a) Profile of grown structure of swollen seed-20% without BZSA in the gel matrix at different UV irradiation times. (b) Magnification of black line (0.5 min UV irradiation) in (a).

7) The idea of fluorescent crosslinker is quite clever. From Figure 2g, we can see that the surface of new grown structure has brightest color, which is almost same as the seed part. If it is due to the transesterification, I think the color should gradually change from bright to dark in the growth direction. To me, it seems that the UV light leads to the local swollen of irradiated region. At same time, the polymerization also happens in this region. Then the polymerization-diffusion cycle leads to the gradual swollen of the network and the new polymerization always happens insides the original network. This process gives the fluorescent images of Figure 2f and 2g.

Response: Thank you for insightful comments! We basically agree with the comment about the growth process. However, the fluorescence change is not simply from bright to dark in the growth direction. As shown in Figure R7 (Figure S20 in the revised manuscript), the fluorescence intensity of the surface of new grown structure is nearly same as that observed in the seed part and then the intensity gradually changes from dark to bright. The brightest top layer indicates that no growth occurs on the surface layer. We attribute it to two possible reasons: 1) monomer molecules in this region is easy to evaporate, especially in photopolymerization state (the temperature in the irradiated area can be up to $>60\text{ }^{\circ}\text{C}$); 2) the “skin layer” on the surface might has a highly crosslinked structure that prevents efficient swelling. The change from dark to bright can be explained by the gradual

swollen of the network. Based on the growth curves (Figure R1a), the growing rate decreases with time due to the consumption of the monomer. Therefore, the diluting effect in fluorescence decreases from top (early stage) to bottom (late stage) in the growth direction.

Figure R7. (a) Swollen seed-20% dyed by PDIDA in the seed after growing. (b) Fluorescence intensity of L1 on (a). (c) Fluorescence intensity of L2 on (a).

These results and discussion have been integrated into the revised manuscript.

8) In Figure S14, the growth height shows a nonmonotonic change with diameter of photomask. Do the authors have some explanations? In Figure 15, did the growth reaches equilibrium when comparing the effect of light energy?

Response: In Figure S14 (Figure S24 in the revised manuscript), with increasing the irradiation diameter, the growth height increases in the range from 266 μm to 600 μm but decrease in the range of $>600 \mu\text{m}$. We attribute the increase to the photopolymerization-induced thermal effect. The temperature of the irradiated region can be up to $>60 \text{ }^\circ\text{C}$ by photothermal effect from polymerization under UV exposure, which would accelerate the diffusion rate of liquid molecules and thus the growth. As considering the thermal conductivity, the thermal effect increases with irradiation diameter. When the diameter becomes larger, the contribution of the thermal effect decreases. As considering thermal dissipation, increasing irradiation diameter favored temperature elevation. On the other hand, increasing diameter also elongated the diffusion distance and thus reduced the

growth. This reducing effect became more obviously in large diameter range ($>600\ \mu\text{m}$).

In Figure S15 (Figure S25 in the revised manuscript), the growth does not reach its equilibrium state yet. We did not collect data in equilibrium state because we want to catch the kinetical state, rather than a thermodynamic state (note that the irradiation time is same but different light density leads to different irradiation energy dose). This experiment was conducted to figure out a basic technique question: what's the minimal light density for triggering the growth.

Reviewer #2 (Remarks to the Author):

Micro and nano patterns are very important to surface chemistry and physicals, and their fabrication technology is being attracted much attention. This manuscript from Lulu Xue, et.al reports a direct photo-induced strategy to control site-specific growth of microstructure from swollen elastomers. Microstructures can grow directly from flat substrates without any preprogramming by guiding continuously molecule transport, interconversion, and reconfiguration sequentially. This is innovative and meaningful for the formation of pattern with spatially controllable, dose-dependent. I suggest a revision of the manuscript before it can be accepted for publication. Some comments shown as following.

Response: Thank for the positive comments! The manuscript has been revised based on the suggestions from the reviewer.

1. How about the minimum resolution of pattern can be fabricated by this method? Is it possible to produce nano or submicrometer scale pattern.

Response: We did not try to produce a submicrometer scale pattern since the minimum resolution of our primo microscopy is $2\ \mu\text{m}$ (it is possible to generate a structure with a diameter of $2\ \mu\text{m}$). Our method might be inefficient to produce a well-defined nano or submicrometer scale pattern since the diffusion of generated counter ions and radicals could distort the regular structures in nano or submicrometer scale.

2. In Figure2(g), the fluorescence weakening in the grown region may be caused by fluorescence quenching by UV-irradiation for long time, rather than significantly diffusion. How to eliminate the factors of fluorescence quenching in this experiment?

Response: Thank for this insightful comment! We do not expect any irradiation-induced fluorescence quenching in the dye monomer under our irradiation condition. To confirm this, we have evaluated the photostability of the dyed sample under UV irradiation (Figure R8 and Figure S19 in the revised manuscript). No fluorescence quenching was observed after UV irradiation for 30 min. Therefore, the fluorescence weakening in the grown region is caused by the transport of liquid components. This part has been added in the revised manuscript.

Figure R8. Fluorescence spectrum of PDI dyed seed-20% thin film before and after UV irradiation.

3. In Figure 2(b), why does the grown height of seed-0% suddenly increase even exceed the height of seed-20% after light off, while the height of seed-20% remains unchanged?

Response: The height of grown structure in seed-0% increase after stored in dark for 32 h but not suddenly (sorry for misleading label in Figure 2b). As for the higher grown structure observed in the NBA-free sample, we emphasize that the structure become bigger during storage in dark because liquid compositions (feed solution) diffuse into the grown structure, rather than because of growth. Note that photopolymerization consume the monomer and crosslinker (liquid compositions) in the irradiated region, inducing a concentration gradient which drives liquid compositions to diffuse into the region (and then further polymerization). Such polymerization-diffusion cycle leads to the growth of the structure. In this cycle, photopolymerization is significantly faster than diffusion (a complete photopolymerization under our irradiation condition is ~2 min while the diffusion time is ~4 hours—see more detail in Figure S5 and S7 in the revised manuscript). Therefore, after the irradiation, generated concentration gradient still exists and continues driving liquid compositions (monomer and crosslinker) to diffuse into the grown structure to make the structure become bigger (swollen state). The obtained structure is irregular and brittle. To confirm this, we washed the grown structure and observed a remarkable shrinkage (washed state, Figure R9). Such irregular structure is undesirable. In the NBA-contained system, carboxyl side chains would be generated (from NBA units) to reduce such diffusion effect and the obtained structure is regular.

We have integrated these results into the revised manuscript to make our claim more convincing and clearer.

Figure R9. Profile of grown structures of seed-0% stored in dark for 32 h before and after washing by ethanol.

4. Page7, it is mentioned that “after the charge disappears, the liquid composites diffuse out from the irradiated region, resulting in a cave surface”, It is better to give more comments on this point.

Response: We have improved our statement in this part to make it clearer in our revised manuscript. Note that upon irradiation, the NBA unit would form ion pair of $-\text{COO}^-\text{H}^+$ at first which turn to $-\text{COOH}$ soon. The ion pair can increase the osmotic pressure but the carboxyl side groups do not. Moreover, the carboxyl side groups weaken the swelling ability of the irradiated region comparing to intact seed-20%. Therefore, the irradiation region changes from a convex (charge pair) to a concave (carboxyl) state.

5. Can the average rate of mass transport be calculated? The specific value?

Response: It is difficult to directly calculate the rate of mass transport during growth since the molecules transport inside the matrix in which the rate of mass transport continues changing. Since the mass transport depends on the diffusion, we evaluated the rate of mass transport in the growth by measuring the diffusion rate of a monomer analogue (4-hydroxybutyl acetate) under irradiation condition. An analogue was used because HBA monomer will undergo polymerization during irradiation even without any initiator. The polymerization will change the composition of the diffusing liquid. To measure the diffusion rate, a fresh seed-20% sample with thickness of 500 μm was immersed into 4-hydroxybutyl acetate and its weight was recorded at different time.

The diffusion rate can be determined using the following equation (Eur. Polym. J. 2002, 38, 2133-2141; React. Funct. Polym. 2007, 67, 865-882):

$$F = \frac{M_t - M_0}{M_0} = Kt^n \quad (1)$$

where F is the rate of diffusion per area; K is a swelling constant, t is the time (s), n is a swelling exponent; M_t and M_0 are the weight of the swollen and dry sample at time t , respectively. From equation (1), we know that

$$\ln F = \ln K + n \ln t \quad (2)$$

We plot $\ln F$ versus $\ln t$ (Figure R10) by using the kinetic of swelling yields straight lines up to

almost 60% increase in the mass of swollen sample (J. Polym. Sci.: Polym. Phys. Ed. 1983, 21, 983-997; Eur. Polym. J. 2000, 36, 2685-2692). The swelling exponents n and the swelling constant K were calculated from the slopes and intercept of the lines. The intercept K value was used for determination of the diffusion coefficient D :

$$K = 4\sqrt{D/\pi r^2} \quad (3)$$

where D is the diffusion coefficient ($\text{cm}^2 \text{s}^{-1}$), r is the radius of the cylindrical seed-20% sample (cm).

Combing with the equation (2), (3) and the plot from Figure R10 (Figure S13 in the revised manuscript), the diffusion coefficient is $4.7 \times 10^{-5} \text{ cm}^2/\text{s}$. As for the control without irradiation, the diffusion coefficient is $4.9 \times 10^{-6} \text{ cm}^2/\text{s}$.

Figure R10. Swelling kinetic curve of cylindrical shaped seed-20% samples under different conditions. Samples with a diameter of 1 cm were used.

6. In this manuscript, it is mentioned that the grown sample retains its shape after stored in dark overnight, how long will it last, and whether there will be collapse or deformation?

Response: The new grown structure from seed-20% is stable. We evaluated the stability of the new grown structure in both swollen and washed states (washing treatment is used to remove unreacted monomer and crosslinker, Figure R11 and Figure S9 in the revised manuscript). After stored for 32 h, the grown structure of as-prepared sample become slightly bigger because liquid diffuses into the grown structure as a result of concentration gradient (note that the change is significantly smaller than that observed in NBA-free sample as shown in Figure R9), which still retained this shape for 1 month in dark. In the washed sample, the grown structure shows nearly same profile even after stored in dark for over 1 months.

Figure R11. Profile of grown structure from swollen seed-20% with (a) and without (b) washing by ethanol before and after stored in dark for 32 h and 1 month. Cylinder structures with diameter of 500 μm could grow up to 250 μm from swollen seed-20%, then one was put in the dark directly, the other was washed by ethanol then put in the dark.

7. How about the quantitative relationship between the expansion rate or cross-linking degree of substrate and the growth height?

Response: We have conducted additional experiments to investigate the relationship between expansion rate/crosslinking degree of the substrate and the height of the grown structure. Actually, the expansion rate of the substrate depends on the crosslinking degree. As shown in Figure R12 (Figure S23 in the revised manuscript), increasing the crosslinking degree decrease the expansion rate within the substrate as well as the height of the grown structure. The high crosslinking degree leads to a rigid substrate and thus limits the expansion rate and the growth.

Figure R12. (a) Strain-stress curve of seed-20% with different fraction of cross-linkers. (b) E-modulus of samples in (a). (c) Swelling ratio of samples of (a) in nutrient solutions. (d) Growth height v.s. different seed-20% samples in (a) under UV light irradiation. Photomask used in (d) with diameter of 500 μm .

Reviewer #3 (Remarks to the Author):

The manuscript presents an elaborate system for inducing spatially and temporally controlled, light-triggered polymerization resulting in a geometrically restricted growth of a polymer sample. The authors propose that the design is based “on the coupling of three kinds of reactions together, including photolysis to generate a driving force for liquid transport, photopolymerization to incorporate the liquid, and transesterification to homogenize the newborn component”.

While the subject matter of the paper is interesting and the amount of the data presented is substantial, the manuscript requires major revision, in order for it to be considered for publication in a high-impact outlet, such as Nature Communications.

Response: We thank for the insightful comments. The manuscript has been carefully revised according to the suggestions.

There are substantial deficiencies in the manuscript, in terms of both the scientific substance, (presented experimental evidence, its interpretation, and conclusions drawn) and the quality of

its presentation.

Response: We have tried our best to improve our manuscript in both presentation and scientific substance. The changes have been highlighted point-by-point in the revised manuscript.

The paper claims that due to the somewhat elaborate design presented (that is the combination of photolysis, liquid transport, polymerization, and transesterification), the authors have developed a “strategy for designing photo-induced growing materials” with “potential applications in transformable materials, additive manufacture, self-healing materials, actuating systems, adaptive materials, etc.” This claim is very broad, while the system presented is quite narrow in scope. The presented strategy could nonetheless be considered a development worthy of publication, but one would still have to ask themselves in what level of a journal. While the level of novelty of this work is debatable and whether it merits publication in a Nature family journal is a matter of opinion, there are several elements in the manuscript that this referee finds to be deficiencies significant enough that without correcting them, the paper cannot be considered for publication.

Response: We claim these potential applications because of the reasons as follows: 1) Transformable materials. The light-control growth allows selectively swelling some domain (as well as shrinking residual domain), which can be used to change material's shape. 2) Additive manufacture. The liquid compositions are integrated inside of the materials without removing surplus material and therefore, it is possible to produce a structure with a size larger than its starting material. 3) Self-healing. Results in Figure 4 have shown this potential. 4) Actuating system. The growing process of a new structure can be utilized as a mechanism to apply force. 5) Adaptive material. Because of the deformability of the materials, the materials can change their shape to adapt the environment during a growing process (the transesterification reaction allows for reconfiguration).

Anyway, we have narrowed our claim to “self-healing materials and rough surfaces” in the revised manuscript.

“We thus envision that its development will benefit areas such as self-healing materials and rough surfaces”

Authors reference a very relevant work by J. Johnson et al (ref. 10: Chen, M. et al. Living additive manufacturing: transformation of parent gels into diversely functionalized daughter gels made possible by visible light photoredox catalysis. ACS Cent. Sci. 3, 124-134 (2017)), but neglect to reference other critically important works from the same group: Zhou, H. X.; Johnson, J. A. Photo-controlled Growth of Telechelic Polymers and End-linked Polymer Gels. Angew. Chem., Int. Ed. 2013, 52, 2235–2238; Gordon, M. B.; French, J. M.; Wagner, N. J.; Kloxin, C. J. Dynamic Bonds in Covalently Crosslinked Polymer Networks for Photoactivated Strengthening and Healing. Adv. Mater. 2015, 27, 8007–8010. This is a serious oversight, because all these papers, taken together, demonstrate that the use of PHOTOACTIVATED steps for growth and post-polymerization modification of polymeric materials for the very similar to this paper's purposes have significant precedent in the literature.

Response: Thank for kindly reminding us these papers! These contributions have been cited and discussed in the revised manuscript.

“Johnson et al. developed a class of growable polymer gels by using trithiocarbonate iniferters as a dynamic connector. The iniferters can incorporate monomer molecules entrapped in the gels to elongate the polymer segments between crosslinked points. Similar idea was applied by Kloxin and co-workers to create covalently crosslinked polymer networks in which crosslinking reactions can be triggered to strengthen the material or heal damage in the material. These examples indicate that growing strategy is promising in post-variation of material’s properties.”

In addition to many problems with the language/grammar used (some of which are listed further down in this review), there are as many instances where the terminology used is simply wrong, often so wrong that it precludes understanding what the authors actually meant. This applies to chemical terminology, materials science terminology, and layman’s language alike. Some examples from the first several pages of the manuscript are below.

Response: We have tried our best to correct and modify all the unsuitable language, grammar and terminology in the revised manuscript.

Abstract:

“The strategy is based on the coupling of three kinds of reactions together, including photolysis to generate a driving force for liquid transport, photopolymerization to incorporate the liquid, and transesterification to homogenize the newborn component.” It is debatable whether photolysis, as such, generates a driving force for liquid transport (see additional discussion of this issue further below). Photopolymerization (or any other polymerization) does not incorporate a liquid, it is a chemical reaction that incorporates a monomer and/or crosslinker into the polymer. Transesterification does not homogenize the newborn component; it results (in this case) in a redistribution of ester linkages within the polymer network(s). And “newborn component” needs a clear definition, in order to be used in a scientific paper.

Response: Sorry for such confusing presentation. The sentence has been changed into *“The strategy is based on the coupling of three kinds of reactions together, including photolysis, photopolymerization, and transesterification. The photolysis is designed to generate ion pair to increase the osmotic pressure driving a nutrient solution containing polymerizable compositions to diffuse into the irradiated region, while the photopolymerization to convert the polymerizable compositions into polymers, and transesterification to homogenize the new-formed and original polymer network structures”* in the revised manuscript.

“We demonstrate its (“light-regulated self-growth”) applications in creating structural surfaces and restoring large damage.” What is meant by “structural surfaces”? What is meant by “large damage”?

Response: The term of “structural surfaces” was used to describe a surface with rough morphology or patterning structure. (Kumar et al., Microstructural and electrical properties of PVP-CH₃COONa

polymer films. *Chem. Sci. Rev. Lett.* **2018**, *7*, 664; Bruggen, B. et al, Structure architecture of micro/nanoscale ZIF-L on a 3D printed membrane for a superhydrophobic and underwater superoleophobic surface. *J. Mater. Chem. A* **2019**, *7*, 2723; Wan, F. et al. Grafting polymer brushes on biomimetic structural surfaces for anti-algae fouling and foul release. *ACS Appl. Mater. Interfaces* **2012**, *4*, 4557–4565). However, we also agree that this term is a special term in computer science. To avoid any misleading, we have changed this presentation, for example, using “making microstructure on surfaces” to replace “creating structural surfaces”.

The term of “large damage” was used to describe a damage with a diameter of 7.5 mm created in self-healable materials, by White et al (White, S.R. et al. Restoration of large damage volumes in polymers. *Science*, 2014, 344, 620-623). Here we used it to describe the surface damage with a diameter of 3.5 mm (Figure 4b).

P. 2: “Johnson et al. developed a class of growable polymer gels containing an initiator of reversible addition-fragmentation chain-transfer polymerization (RAFT) as a connector.” The sentence utilizes chemical terminology, but one cannot refer to RAFT polymerization as “a connector”.

Response: We agree that RAFT as a connector is not a suitable chemical terminology. Herein we have changed this sentence to “Johnson et al. developed a class of growable polymer gels by using trithiocarbonate initiators as a dynamic connector”. This part has been discussed in more detail in the main text in the revised manuscript.

P. 2: “Many strategies have been developed to selectively trigger chemical reactions for spatial functionalization, including light, strength, temperature, moisture, etc.” None of the words used (light, strength, temperature, moisture) can be described as a “strategy”.

Response: Thank you! We have changed “strategies” to “stimuli” in the revised manuscript.

P. 3: “Herein, we report a photo-regulated strategy to control site-specific growth of microstructure from swollen elastomers.” “Site-specific” is not a good word in this context. Likewise, “growth of microstructure from swollen elastomers” is difficult to understand for a polymer scientist and a general reader alike.

Response: The term of “site-specific” is frequently used to describe the capability of light to spatially modulate the properties of a substrate (Shadish, J. et al. Bioactive **site-specifically** modified proteins for 4D patterning of gel biomaterials. *Nat. Mater.* 2019, 18, 1005-1014; Gawade, P. et al. Logic-based delivery of **site-specifically** modified proteins from environmentally responsive hydrogel biomaterials. *Adv. Mater.* 2019, 31, 1902462; Kitayama, Y. et al. Efficient pathway for preparing hollow particles: **site-specific** crosslinking of spherical polymer particles with photoresponsive groups that play a dual role in shell crosslinking and core Shielding. *Langmuir* 2016, 32, 9245-9253; Bio, M. et al. **Site-specific** and far-red-light-activatable prodrug of combretastatin A-4 using photo-unclick chemistry. *J. Med. Chem.* 2013, 56, 3936-3942; Yamaguchi, S. et al. Light-activated gene expression from **site-specific** caged DNA with a biotinylated photolabile protection group. *Chem. Commun.*, 2010, 46, 2244-2246; Jiang, M. et al. **Site-specific**

prodrug release using visible light. J. Am. Chem. Soc. 2008, 130, 4236-4237). In our manuscript, we use light to regulate the growth of microstructure from the surface of swollen elastomers. Since the grown structure only form on the irradiated area, we thus used this term to describe such feature.

We agree that “growth of microstructure from swollen elastomers” is difficult to understand. It has been changed to “*growth from dynamic swollen substrates for making rough surfaces*” in the revised manuscript.

P. 3: “In our design, three kinds of reactions, including photolysis, photopolymerization, and transesterification, are coupled together to guide continuously molecule transport, interconversion, and reconfiguration sequentially.” Even if one can accept (with difficulty) the term “molecule transport”, “molecule ... interconversion and reconfiguration” does not carry much meaning, and “continuously” and “sequentially” only additionally muddle the intended message.

Response: We have changed the presentation to “*In our design, three kinds of reactions, including photolysis, photopolymerization, and transesterification, were coupled together to guide transport of liquid compositions entrapped in the substrates, to convert the polymerizable compositions in the liquids to polymers, and to reconfigure new-formed and original polymers, respectively*”.

P. 4: “while NBA is a photolabile monomer (promoter) that can generate a charge pair of carboxyl ionic and hydrion, to raise high osmotic pressure.” Probably “carboxyl ion” (not “ionic”) is meant, but even if so, what does “to raise high osmotic pressure” mean? Also, I do not remember when last time I saw the term “hydrion” in a scientific/chemistry paper.

Response: Sorry for the wrong terms! The sentence has been changed to “*while NBA is a photolabile monomer (promoter) that can generate an ion pair of $-COO^-H^+$ for inducing an increase in osmotic pressure*” in the revised manuscript. The osmotic pressure of the irradiated region is higher than that of intact one. This pressure difference drives liquid components to diffuse into the irradiated areas.

As stated above, this is just a handful of examples from the first several pages, but there are many more throughout the text. Such misuse of scientific and non-scientific terminology precludes, in many cases, understanding of what the authors actually meant to say, which is unacceptable for a scientific publication in any journal.

Response: We have modified the full text of the manuscript to correct all the unsuitable sentences and terminologies.

Important scientific issues to consider:

1) One of the central messages the authors are making is about the role of photolysis in inducing (stimulating or accelerating, rather) liquid transport from the bulk of the swollen samples into

the areas where irradiation occurred. The authors attribute it to the increase in osmotic pressure. This is a key aspect, by which they distinguish their work from a very carefully done work coming from groups of Matyjaszewski, Kowalewski and Balazs (ref. 23 in the manuscript: Cuthbert, J. et al. Transformable materials: structurally tailored and engineered macromolecular (STEM) gels by controlled radical polymerization. Macromolecules 51, 3808-3817 (2018)). To support their statement, the authors cite ref. 26 (Alexander, S. et al. Charge renormalization, osmotic-pressure, and bulk modulus of colloidal crystals-theory. J. Chem. Phys. 80, 5776-5781 (1984)). This work, however, applies to a substantially different system and if the authors somehow conclude that there is direct relevance between the physical treatment of colloidal crystals presented in this work and their case of inducing faster/more pronounced liquid influx into the irradiated areas of the polymer network, they should explicitly describe and justify the connection. Or explicitly state that the high osmotic pressure argument they are making is a mere hypothesis. It is also quite important to note that the infiltrating liquid in question is not very polar, so one has to reconcile that with the proposed increase in driving force for it to diffuse into the irradiated regions containing charge pairs. There may be other explanations, or hypotheses, compatible with the experimental observations. For example, there are mechanical properties/diffusion/crosslinking arguments to consider.

Response: We agree that the ref. 26 is not the most appropriate reference to support our presentation. In this paper, the authors described how the charge renormalization influenced the osmotic pressure. We used this to support the claim that the generation of ion pair would induce an increase in osmotic pressure. The material systems are indeed far away from our polymer systems. Therefore, a more appropriate paper about using polyelectrolytes to enhance polymer swelling ability to nonpolar organic solvents is cited (Ono, T. et al. Lipophilic polyelectrolyte gels as super-absorbent polymers for nonpolar organic solvents. Nat. Mater 2007, 6, 429–433).

Osmotic pressure is not a simple hypothesis in our system. We have provided solid data to prove the formation of ion pair ($-\text{COO}^-\text{H}^+$) and its contribution to the osmotic pressure of the matrix. Firstly, we have used a control experiment to confirm that charge structures indeed forms when the NBA unit is irradiated (Figure 2c). We have then confirmed that upon the irradiation, swollen indeed happens in the irradiated region. Introducing charge structure in a polymer matrix can lead to a remarkable increase in swelling ratio to different solvents, including polar and nonpolar solvents (Ono, T. et al. Lipophilic polyelectrolyte gels as super-absorbent polymers for nonpolar organic solvents. Nat. Mater 2007, 6, 429–433).

We used non-polymerizable liquids to confirm the contribution of osmotic pressure. In this control experiment (Figure 2d), only the photolysis reaction of NBA units occurs. In this reaction, intermediate ion pair of $-\text{COO}^-\text{H}^+$ forms and finally convert into carboxyl product. As expected, during irradiation, the irradiated region swells, indicating that liquid compositions diffuse into the irradiated region, while after irradiation, the irradiated region shrinks, implying opposite diffusion behavior. This perfect parallel support our hypothesis. Note that although only a change of $\sim 5 \mu\text{m}$ was observed, such difference can be amplified in the polymerization-diffusion cycle.

2) Also, related to this, the attribution by the authors of the less pronounced swelling after sample irradiation (with the promoter present) to the formation of acrylic acid (Fig. S5) is not sufficiently substantiated at all. The evidence of the water contact angle decrease (Fig. S4 a, b) is indirect, while what is claimed as the FTIR evidence of the appearance of carboxyl group (Fig. S4c) and the text accompanying it are less than convincing (both in the substance and in the language used):

Section 6 of SI:

“Water contact angle was used to monitor the photolysis of the substrates. It changes from 95o to 63o after UV irradiation to generate acrylic side group (Figure S4a and 4b). FTIR shows the disappearance of the -NO₂ after UV illumination as the peak at 1528 cm⁻¹ of OH group became broader (Figure S4c).

The photolysis of NBA units to generate acrylic side groups decreases the swelling ratio of the seeds to the nutrient solution. Figure S5 shows the swelling kinetics of the photolytic film (1.4 mm) and the equilibrium swelling ratio decreases from 4.6 to 3.5.”

With the authors’ text and punctuation kept, it seems that the peak at 1528 cm⁻¹ is attributed to OH and it has become broader. This is total confusion. The 1528 cm⁻¹ peak is assigned by the authors to the NO₂ group and the broadening of the OH group signal is i) extremely difficult to see and ii) even if it is there, it is not even a weak evidence of the appearance of a carboxyl group. Also, what is “acrylic side groups”??

Also, though the assignments of the IR bands due to the NO₂ group are likely correct, the IR spectrum of the polymer lacking the nitrobenzene group is needed/desired, in order to give more weight to this assignment.

Response: We agree that the data of water contact angle is not the direct evidence of the formation of carboxyl side chain from NBA unit. However, it is quite evident that the NBA units undergoes photolysis to generate carboxyl side chains because of the reasons as follows: 1) It is well-known that NBA is a photolabile molecule in which the 2-nitrobenzyl group can be removed by UV light (Adv. Funct. Mater. 2008, 18, 1501–1508; J. Am. Chem. Soc. 2009, 131, 13315-13319; J. Mater. Chem., 2010, 20, 8920-8926). 2) We also provide solid data (Figure S11) to prove that the NBA prepared in our hands can undergo photolysis reaction in our irradiation condition. 3) In order to further confirm the photolysis reaction in film state, we have compared the FTIR spectra of the sample before and after UV light irradiation (Figure S4). The peaks assigned to the –NO₂ group (asymmetric NO₂ stretch at 1528 cm⁻¹ and symmetric NO₂ stretch at 1343 cm⁻¹) disappear after irradiation.

We agree that the broadening of the OH group signal is extremely difficult to see in our system. The statement has been deleted.

The wrong presentation of “FTIR shows the disappearance of the -NO₂ after UV illumination as the peak at 1528 cm⁻¹ of OH group became broader (Figure S4c)” has been changed to “*In FTIR spectra, the peaks assigned to the group of -NO₂ (1528 cm⁻¹ and 1343 cm⁻¹) disappeared after UV irradiation”.*

The term of “acrylic side groups” is not appropriate here and it has been changed to “carboxyl side groups” in the revised manuscript.

3) P. 5: *The authors state “In company with the polymerization, thermal effect generated by polymerization would trigger transesterification reactions to release any polymerization-induced mechanical tension in such dynamic networks.27, 28”*

There is no evidence presented of the temperature increase due to the polymerization. For this statement to be made, an IR-camera measurements are necessary. Moreover, the references 27 and 28 invoke temperatures of 130 C and higher for the transesterification reactions as the redistribution mechanism in the polymer networks they describe. So, here, as in the case of the increased osmotic pressure explanation (hypothesis, rather), the used references do not offer direct support to the statements made by the authors.

Response: Thank for this insightful comment! The temperature for efficient transesterification reaction depends on the catalyst. We have selected a catalyst (benzenesulfonic acid) which can trigger transesterification under mild conditions (a catalyst temperature of 55 °C was reported by Self, J. et al. in “Brønsted-acid-catalyzed exchange in polyester dynamic covalent networks. ACS Macro Lett. 2018, 7, 817–821”). According to reviewer’s comment, we have measured the temperature of the gel matrix (swollen seed-20%) under irradiation (Figure R13 and Figure S14 in the revised manuscript). It was found that under our irradiation condition, the temperature can increase from 25 °C to 62 °C after 1 min UV irradiation. This temperature is high enough for the transesterification reaction.

Figure R13. Infrared-camera image of swollen seed-20% before (a) and after (b) 1 min UV irradiation. White dotted zone shows the position of swollen sample, while black dotted zone stands for the light irradiation area.

4) Fig. 2b:

What is the explanation of the seemingly very high growth of the control sample (the last point of the magenta curve)? The accompanying text (big part of P.6) leaves, again, a strong impression of unsubstantiated hypothesizing, especially when coupled with a frivolous use of several different tenses, as the excerpt below shows.

“Compared to this, the control also shows a growth in the irradiated region. It was explained (WHEN?? BY WHOM??) by the fact that photopolymerization consumes the nutrient solution to form new polymer networks in the irradiated region, which would create a nutrient gradient to drive the liquid compositions to diffuse into the irradiated region to join the polymerization. Such diffusion-polymerization cycle thus led to the growth. However, this promoter-free growth is significantly slower than the promoter-presenting growth and the obtained structure is also remarkably smaller. Moreover, the grown structure seriously distorts after stored in dark. We attributed this to the fact that the growth kinetics (~ 50 min) is significantly faster than the diffusion kinetics of the nutrient (~ 12 hours).”

To use the term “kinetics” (rather than “time scale”, for example), in the absence of kinetic measurements, and to measure it in minutes and hours is another example of careless use of scientific terminology.

Response: For the higher grown structure observed in the NBA-free sample, we emphasize that the structure become bigger during storage in dark because liquid compositions (nutrient solution) diffuse into the grown structure, rather than because of growth. Note that photopolymerization consume the monomer and crosslinker (liquid compositions) in the irradiated region, inducing a concentration gradient which drives liquid compositions to diffuse into the region (and then further polymerization). Such polymerization-diffusion cycle leads to the growth of the structure. In this cycle, photopolymerization is significantly faster than diffusion (a complete photopolymerization under our irradiation condition is ~2 min while the diffusion time is ~4 hours—see more detail in Figure S5 and S7 in the revised manuscript). Therefore, after the irradiation, generated concentration gradient still exists and continues driving liquid compositions (monomer and crosslinker) to diffuse into the grown structure to make the structure become bigger (swollen state). The obtained structure is irregular and brittle. To confirm this, we washed the grown structure and observed a remarkable shrinkage (washed state, Figure R14 and Figure S8 in the revised manuscript). Such irregular structure is undesirable. In the NBA-contained system, carboxyl side chains would be generated (from NBA units) to reduce such diffusion effect and the obtained structure is regular.

We have integrated these results into the revised manuscript to make our claim more convincing and clearer.

Figure R14. Profile of grown structures of seed-0% stored in dark for 32 h before and after washing by ethanol.

For the different tenses, we have tried our best to correct and modify them in our revised manuscript.

As for using the term of “kinetic” to show the time scale, we are sorry for the misleading presentation. The growth reaches a plateau when most of monomer and crosslinker around the irradiated region have been consumed; here the time for reaching this is defined as growth time (“growth kinetics”). While the time for reaching completely swelling of seed-20% sample is defined as diffusion time (“diffusion kinetic”). We have modified these in our revised manuscript, respectively.

5) If/when the above comments to the substance are taken into account and a major revision to the paper is made, as importantly, the English grammar and word usage of the text will need significant work. The examples are too numerous to list. Here is just a selection:

“...following by transport inside...” – followed by...

“fundamental different”

“fabricate artificial substances”

“since the lack...” (grammar)

“to apply growing method to create different structure”

“among them light takes advantages of environment friend, noncontact...”

“to character its photostability”

“a seed exhibit”

“a regularly structure tardily grow up from...”

“growing kinetics”

“benzenesulfonic acid” – benzenesulfonic acid

“Irgacue” - Irgacure

Response: We have checked carefully and tried our best to modify all the unsuitable terminologies in the revised manuscript.

Also, BSA is a widely used abbreviation that means “bovine serum albumin”. Here it is used to mean Benzenesulfonic acid; maybe to avoid this abbreviation altogether?

Response: We have changed it to “BZSA” all over the revised manuscript.

In conclusion, though the material presented in the manuscript is interesting and it can attract a multidisciplinary audience, in the current form, the manuscript reads more as a poorly written collection of data lacking careful and well-supported discussion and interpretation. To claim the novelty, as it is claimed now, the distinctions of the current work from the previously published relevant works need to be made in a scientifically more profound and clear way. Where there is not enough experimental proof and substantiation from published data, the presented explanations should be left as hypotheses. Without such a major revision, the manuscript cannot be recommended for publication, in the opinion of this reviewer.

Response: We thank the insightful comments which immediately help us to improve our manuscript. Now additional data have been collected to strengthen our statement and the manuscript has been revised carefully. We believe that the revised manuscript with addressing all the comments from referees accomplishes the requisite of generality and broad interest for the community of Nature Communications.

Reviewers' comments:

Reviewer #1 (Remarks to the Author):

The authors have addressed most of my concerns and now I recommend this paper for publication in Nature Communications.

Reviewer #2 (Remarks to the Author):

This work is on a very hot topic of self-growth of materials. Micro and nano patterns are very important to surface chemistry and physicals, and their fabrication technology is being attracted much attention. This work provides a useful method for surface modification or surface pattern of gel materials.

Reviewer #3 (Remarks to the Author):

I will restrict this review to very few key items that have formed my opinion that the paper is not of the level appropriate for publication in Nature Communications. I feel that while the subject matter and the presented system are interesting, the level of novelty, data interpretation, and the quality of presentation are not at the level appropriate for this Journal. The work the Authors did on revising the manuscript and addressing the reviewers' comments, including my own, failed to sway me towards recommending it for publication.

1) Much of my comments regarding the language remain unaddressed. New text in the manuscript suffers in many places from the same deficiencies as the initial one, as does the newly-crafted rebuttal letter, as did and does the text of Supplementary Material. I listed some examples in my previous review, but there were plenty more, which I did not list, and now new ones have appeared in the revision. These deficiencies alone make the manuscript not acceptable for publication, in my opinion. This time I am choosing not to provide a partial list. I did this last time in hopes that the paper would undergo a critical editing of the language and terminology used. The work the Authors have done in this regard fell short of my expectations.

2) The paper is too heavy on hypotheses that the Authors present to look as confirmed facts. To their credit, they report on a number of control experiments that they did, attempting to justify their hypotheses. Some of the theories they put forward are probably correct, but there is a central one - relating to the irradiation-generated osmotic pressure as the driving force for mass transport - which remains problematic, in my opinion. Describing the previous work by others, the Authors state upfront: "However, in these previous examples, photo-induced reactions were utilized to convert monomer/crosslinker to polymers, rather than to guide mass transport." This is, in their own opinion, the main scientific advance they make in this work, compared to the previously reported work by other researchers. This contribution is important, though as I stated in my previous review, this level of novelty may not justify its publication in a Nature family journal, in my opinion.

The Authors consistently stress that the material influx into the irradiated regions is due to the increased osmotic pressure. As I stated in my previous review, this is a hypothesis that is poorly substantiated. The previously used reference (removed and replaced in the current version) described a system very far from the one used by the Authors, while newly introduced reference #26 deals with systems containing strongly lipophilic anions and cations, not carboxyl and proton, and uses the term "osmotic pressure" solely in the context of water absorbency by polyelectrolytes - NOT while describing influx of organic solvents into organic gels. I maintain that the osmotic pressure argument advocated by the Authors is not adequately supported and other mechanisms may be consistent with the effects observed - as I already stated when reviewing the paper first time. Notably, another reviewer raised a similar concern. Additionally, The Authors observe a temperature increase on irradiation and they attribute it to the exothermicity of polymerization. The experiment lacks an important control, in which no polymerization takes place, but UV

irradiation is switched on. Is there/would there be a temperature increase in this control? If yes, then mechanical softening factor (which could/would facilitate liquid transport to softer regions) is not out of question. Even if no such temperature increase takes place, does the polymer get softer due to a possible UV-triggered partial chain scission? Is there a convincing evidence that these factors are not at play? In short, I was not convinced with the osmotic pressure argument first time around and remain unconvinced now. And while this is claimed to be the key feature of the reported system that sets it apart from the existing research, the unconvincing interpretation additionally takes away from the impact of the manuscript and, therefore, from its suitability for publication in this Journal.

3) The descriptions of carboxyl and proton generation and their ultimate recombination are another example of how some of the Authors' hypotheses (even if there is a reason behind them) are taken a little too far and implied too strongly in substantiation of the observed experimental facts. There is no data in the manuscript that demonstrates/determines the extent/magnitude to which these ion pairs are generated, their lifetimes and the like, while broad conclusions are reached that rely on the (yet unknown) lifetimes of these species. I mention this simply to illustrate the point that extensive unsubstantiated theorizing takes away from quite interesting experimental data that the manuscript reports on.

4) Certain parts of the rebuttal letter left me a bit perplexed. For example, I commented on the term "site-specific" as not the best one in the context the Authors were (and still are) using it (as they are still using many cumbersome phrases and poor terminology that I did not explicitly mention in my first review). Again, in my opinion, it was not and remains not a good choice of words to describe spatial control over where the monomer/cross-linker are transported to and from where the microstructure growth takes place as a result. The term "site-specific" is used in chemical literature almost exclusively to define a site within a molecule, even if within a macromolecule, but not to describe an area on a macroscopic object. The examples of its use in the literature that the Authors list in their rebuttal simply confirm this.

In summary, in the opinion of this reviewer, the revised manuscript is not of the level and of the presentation quality that the readership of Nature Communications expects.

Point-by-Point Response to Referees

Reviewer #1 (Remarks to the Author):

The authors have addressed most of my concerns and now I recommend this paper for publication in Nature Communications.

Response: We appreciate very much for the reviewer's input and suggestions to the manuscript.

Reviewer #2 (Remarks to the Author):

This work is on a very hot topic of self-growth of materials. Micro and nano patterns are very important to surface chemistry and physicals, and their fabrication technology is being attracted much attention. This work provides a useful method for surface modification or surface pattern of gel materials.

Response: We thank very much for the reviewer's insightful comments to the manuscript.

Reviewer #3 (Remarks to the Author):

1) Much of my comments regarding the language remain unaddressed. New text in the manuscript suffers in many places from the same deficiencies as the initial one, as does the newly-crafted rebuttal letter, as did and does the text of Supplementary Material. I listed some examples in my previous review, but there were plenty more, which I did not list, and now new ones have appeared in the revision. These deficiencies alone make the manuscript not acceptable for publication, in my opinion. This time I am choosing not to provide a partial list. I did this last time in hopes that the paper would undergo a critical editing of the language and terminology used. The work the Authors have done in this regard fell short of my expectations.

Response: Although we are not native English speakers, we have invited a native English speaker to check the language. Basically, we used past tense to state what we did but present tense to describe our results.

2) The paper is too heavy on hypotheses that the Authors present to look as confirmed facts. To their credit, they report on a number of control experiments that they did, attempting to justify their hypotheses. Some of the theories they put forward are probably correct, but there is a central one - relating to the irradiation-generated osmotic pressure as the driving force for mass transport - which remains problematic, in my opinion. Describing the previous work by others, the Authors state upfront: "However, in these previous examples, photo-induced reactions were utilized to convert monomer/crosslinker to polymers, rather than to guide mass transport." This is, in their own opinion, the main scientific advance they make in this work, compared to the previously reported work by other researchers. This contribution is important, though as I stated in my previous review, this level of novelty may not justify its publication in a Nature family journal, in my opinion.

Response: The novelty of our manuscript is not only “to guide mass transport” but also the combination of three kinds of reactions together to get spatially-controlled growth from a flat surface. The statement in the second paragraph mainly focuses on the stimulus of “light”. Actually, guiding mass transport in a closed system by light is crucial but has not been discussed yet in previous reports. In addition to the light-control mass transport (it is only one step of the growth), the coupling of transesterification, the concept of the polymerization-diffusion growth cycle, and the outcome of the growth of well-defined microstructure on surfaces are all novel in this emerging field of growing materials.

The Authors consistently stress that the material influx into the irradiated regions is due to the increased osmotic pressure. As I stated in my previous review, this is a hypothesis that is poorly substantiated. The previously used reference (removed and replaced in the current version) described a system very far from the one used by the Authors, while newly introduced reference #26 deals with systems containing strongly lipophilic anions and cations, not carboxyl and proton, and uses the term "osmotic pressure" solely in the context of water absorbency by polyelectrolytes - NOT while describing influx of organic solvents into organic gels. I maintain that the osmotic pressure argument advocated by the Authors is not adequately supported and other mechanisms may be consistent with the effects observed - as I already stated when reviewing the paper first time. Notably, another reviewer raised a similar concern. Additionally, The Authors observe a temperature increase on irradiation and they attribute it to the exothermicity of polymerization. The experiment lacks an important control, in which no polymerization takes place, but UV irradiation is switched on. Is there/would there be a temperature increase in this control? If yes, then mechanical softening factor (which could/would facilitate liquid transport to softer regions) is not out of question. Even if no such temperature increase takes place, does the polymer get softer due to a possible UV-triggered partial chain scission? Is there a convincing evidence that these factors are not at play? In short, I was not convinced with the osmotic pressure argument first time around and remain unconvinced now. And while this is claimed to be the key feature of the reported system that sets it apart from the existing research, the unconvincing interpretation additionally takes away from the impact of the manuscript and, therefore, from its suitability for publication in this Journal.

Response: We agree that the term “osmotic pressure” is not appropriate here to describe the swelling behavior of ionic organic gels. It has been suggested that the swelling behavior of the ionic organic gels is controlled primarily by the compatibility of the polymer chains with the media, and incorporation of the dissociable ionic groups can enhance the swelling abilities in more polar compatible solvents (*Nat. Mater.*, 2007, 6, 429-433). The comparison results of photolysis-absent and -present growths (also NBA-contained and NBA-free systems in growth experiments) indicate that more mass transports into the irradiated region. In photolysis-absent system, the driving force for mass transport is the concentration gradient (the same driving force for swelling). The more mass moved into the irradiated region in photolysis-present growth during the same growth time indicated that the photolytic reaction of NBA units should generate an extra driving force to accelerate the mass transport. Since the final photolytic products of NBA units weaken (rather than enhance) the swelling ability, we paid our attention to the intermediates generated in the photolytic reaction. Therefore, we designed the experiment described in Figure 2d. The results (the irradiated

region swells at first and shrinks after the sample was stored in dark to reach an equilibrium state) support our hypothesis that unstable intermediates provide extra effect to enhance the swelling ability. The photolytic reaction of NBA units is well known, which generates two moieties: carboxyl moiety and the products of *o*-nitrobenzyl moiety. Based on this understanding, we assume the formation of dissociable ionic group of -COO^- . The linear control polymer was thus designed to confirm this assumption. As expected, a negative Zeta potential (a negative charge polymer species) obtained in the photolysis of this linear control polymer (Figure 2c). Additional control experiment of the photolysis of 2-nitrobenzyl alcohol (which is suggested to generate similar intermediate of the *o*-nitrobenzyl moiety of NBA unit, see answer to Question 3) also supports that the intermediate of carboxyl moiety, most probably the dissociable ionic group of -COO^- , enhances the swelling ability. The dissociable -COO^- groups along the polymer chains would induce electrostatic repulsion to make polymer segments to expand and thus enhance the swelling ability (same driving force in both hydrophilic and lipophilic polyelectrolyte gels). Note that the monomer (alcohol) used in our system is quite polar (hexanol is a polar solvent showing a polarity of 13). Therefore, it should have good compatibility with the group of -COO^- . In addition, all of other data (Figure S7 and Figure S13) also support this enhanced swelling ability.

Although the cited reference (*Nat. Mater.*, 2007, 6, 429-433) do not describe the same groups (to our best knowledge, it is the first time to discuss the charge photolytic intermediate of NBA-like structure), the principle to enhance the swelling ability of organic gel was discussed in detail in this paper. We thus believe it is an appropriate reference here. Moreover, since it has been commented that “it is also quite important to note that the infiltrating liquid in question is not very polar, so one has to reconcile that with the proposed increase in driving force for it to diffuse into the irradiated regions containing charge pairs”, we believe that it might be a general confusion. This reference could help reader to get the important conclusion that one can enhance the swelling ability of a substrate to nonpolar solvent by introducing dissociable ionic groups.

According to the reviewer’s suggestion, we have conducted a control experiment of recording temperature changing in a swollen seed-20% containing non-polymerizable liquids. The sample was prepared by immersing seed-20% in a solution containing 4-hydroxybutyl acetate, I-819, and BZSA. As shown in Figure R1’, our UV irradiation condition would not induce an increase in temperature by itself.

Figure R1’. Infrared camera image of swollen seed-20% containing non-polymerizable liquids under different conditions: (a) before irradiation, (b) after 1 min irradiation, (c) after 60 min irradiation. A 365 nm LED lamp (10 mW/cm^2) was used for irradiation. White dotted zone shows the position of swollen sample, while black dotted zone stands for the light irradiation area.

We agree with the concern of possible UV-triggered partial chain scission. Four kinds of control samples were subjected to UV irradiation to study the possibility of UV-triggered partial chain

scission: seed-20%, swollen seed-20% containing non-polymerizable liquids, seed-0%, and swollen seed-0% containing non-polymerizable liquids. The former two contain photoresponsive NBA units while the latter two do not. As shown in Figure R2', after 30 min UV irradiation, the compression E-moduli of samples with NBA units (seed-20% sample and swollen seed-20% containing non-polymerizable liquids) slightly increase while those of non-photoresponsive samples (seed-0% sample and swollen seed-0% containing non-polymerizable liquids) do not change. These results indicated that UV-triggered partial chain scission could be neglect in our irradiation process.

Figure R2'. Compression E-moduli of different samples before and after UV irradiation for 30 min. (a) Samples with NBA units. (b) Samples without NBA units. A 365 nm LED lamp (10 mW/cm²) was used for irradiation. Non-polymerizable liquids contain 4-hydroxybutyl acetate, I-819, and BZSA.

3) The descriptions of carboxyl and proton generation and their ultimate recombination are another example of how some of the Authors' hypotheses (even if there is a reason behind them) are taken a little too far and implied too strongly in substantiation of the observed experimental facts. There is no data in the manuscript that demonstrates/determines the extent/magnitude to which these ion pairs are generated, their lifetimes and the like, while broad conclusions are reached that rely on the (yet unknown) lifetimes of these species. I mention this simply to illustrate the point that extensive unsubstantiated theorizing takes away from quite interesting experimental data that the manuscript reports on.

Response: We attributed the formation of the charge structure to the generation of dissociable ionic group of -COO⁻ based on the following consideration: the result of Figure 2c indicates the formation of a kind of negative charge species (with positive counterion) which would recombine, during the photolysis of NBA units. Since the NBA unit generates both carboxyl and *o*-nitrobenzyl moieties, we agree that there is another possibility to generate negative charge species from *o*-nitrobenzyl moiety. To probe this possibility, we selected 2-nitrobenzyl alcohol as a control since it would undergo similar photolytic reaction as NBA unit but does not generate carboxyl group (J. Photoch., 1987, 36, 85-97; J. Am. Chem. Soc. 1991, 113, 4303-4313; J. Am. Chem. Soc. 2004, 126, 4581-4595). As shown in Figure R3', photolysis of 2-nitrobenzyl alcohol does not induce any obvious change in the Zeta potential, indicating that the change in Figure 2c should be attributed to the generation of -COO⁻. Although we did not know the lifetime and magnitude of the -COO⁻ in molecular level, their sums are clear: intensity of ~-32 mV and lifetime of 12 min.

Figure R3'. Zeta potential of 2-nitrobenzyl alcohol at different irradiation times.

4) *Certain parts of the rebuttal letter left me a bit perplexed. For example, I commented on the term "site-specific" as not the best one in the context the Authors were (and still are) using it (as they are still using many cumbersome phrases and poor terminology that I did not explicitly mention in my first review). Again, in my opinion, it was not and remains not a good choice of words to describe spatial control over where the monomer/cross-linker are transported to and from where the microstructure growth takes place as a result. The term "site-specific" is used in chemical literature almost exclusively to define a site within a molecule, even if within a macromolecule, but not to describe an area on a macroscopic object. The examples of its use in the literature that the Authors list in their rebuttal simply confirm this.*

Response: "Site-specific" is widely used to describe localized functionalization of materials by light from nano- to micro-levels (Shadish, J. et al. Bioactive site-specifically modified proteins for 4D patterning of gel biomaterials. *Nat. Mater.* 2019, 18, 1005-1014; Jiang, M. et al. Site-specific prodrug release using visible light. *J. Am. Chem. Soc.* 2008, 130, 4236-4237). In our manuscript, we used light to regulate the growth of microstructure from the surface of swollen elastomers. To further avoid any misleading, we have changed "site-specific" to "localized".

REVIEWERS' COMMENTS:

Reviewer #1 (Remarks to the Author):

I think the questions raised by referee #3 do exist and the authors also paid efforts to address them. As the authors said, they coupled three kinds of reactions together, which is rather complicated. Therefore, it is not easy to make everything clear. But I think the most important thing is that the concept the authors proposed is new and the results are interesting. The referee #3 mainly concerned about the explanation of the results, and he/she think it is based on a lot of hypotheses. I suggest the authors to indicate all these hypotheses clearly in the manuscript, which may address the referee #3's concerns. In my opinion, this paper can be accepted after this suggestion is adopted.

Reviewer #2 (Remarks to the Author):

I think that the authors have addressed technical concerns of reviewer #3, and recommend this paper for publication.

Point-by-Point Response to Referees

Reviewer #1 (Remarks to the Author):

I think the questions raised by referee #3 do exist and the authors also paid efforts to address them. As the authors said, they coupled three kinds of reactions together, which is rather complicated. Therefore, it is not easy to make everything clear. But I think the most important thing is that the concept the authors proposed is new and the results are interesting. The referee #3 mainly concerned about the explanation of the results, and he/she think it is based on a lot of hypotheses. I suggest the authors to indicate all these hypotheses clearly in the manuscript, which may address the referee #3's concerns. In my opinion, this paper can be accepted after this suggestion is adopted.

Response: We appreciate very much for the reviewer's suggestions. We have made our presentation clearer in our revised manuscript.

Reviewer #2 (Remarks to the Author):

I think that the authors have addressed technical concerns of reviewer #3, and recommend this paper for publication.

Response: We thank very much for the reviewer's comments to the manuscript.